# SMOC-1 interacts with both BMP and glypican to regulate BMP signaling in *C. elegans*

**Melisa S. DeGroot, Byron Williams, Timothy Y. Chang, Maria L. Maas Gamboa, Isabel M. Larus, Garam Hong, J. Christopher Fromme, Jun Liu** *

Department of Molecular Biology and Genetics, Cornell University, Ithaca, New York, United States of America

* kelly.jun.liu@cornell.edu

**Data Availability Statement:** All relevant data are within the paper and its Supporting Information files.

## Abstract

Secreted modular calcium-binding proteins (SMOCs) are conserved matricellular proteins found in organisms from *Caenorhabditis elegans* to humans. SMOC homologs characteristically contain 1 or 2 extracellular calcium-binding (EC) domain(s) and 1 or 2 thyroglobulin type-1 (TY) domain(s). SMOC proteins in *Drosophila* and *Xenopus* have been found to interact with cell surface heparan sulfate proteoglycans (HSPGs) to exert both positive and negative influences on the conserved bone morphogenetic protein (BMP) signaling pathway. In this study, we used a combination of biochemical, structural modeling, and molecular genetic approaches to dissect the functions of the sole SMOC protein in *C. elegans*. We showed that CeSMOC-1 binds to the heparin sulfate proteoglycan GPC3 homolog LON-2/glypican, as well as the mature domain of the BMP2/4 homolog DBL-1. Moreover, CeSMOC-1 can simultaneously bind LON-2/glypican and DBL-1/BMP. The interaction between CeSMOC-1 and LON-2/glypican is mediated specifically by the EC domain of CeSMOC-1, while the full interaction between CeSMOC-1 and DBL-1/BMP requires full-length CeSMOC-1. We provide both in vitro biochemical and in vivo functional evidence demonstrating that CeSMOC-1 functions both negatively in a LON-2/glypican-dependent manner and positively in a DBL-1/BMP-dependent manner to regulate BMP signaling. We further showed that in silico, *Drosophila* and vertebrate SMOC proteins can also bind to mature BMP dimers. Our work provides a mechanistic basis for how the evolutionarily conserved SMOC proteins regulate BMP signaling.

## Introduction

The highly conserved bone morphogenetic protein (BMP) pathway is used in a variety of developmental and homeostatic processes across metazoans [1]. BMPs are members of the transforming growth factor β (TGFβ) superfamily, and BMP signaling is activated upon binding of secreted BMP ligands to complexes of the type I and type II BMP serine/threonine receptor kinases, which results in the intracellular phosphorylation of receptor-activated R-Smads. Once phosphorylated, the activated R-Smads complex with common mediator Smads (Co-Smads) and enter the nucleus to regulate downstream gene transcription.

**Funding:** This work was supported by R35 GM136258 to JCF and R35 GM130351 to JL. MSD was partially supported by a National Science Foundation (NSF) Graduate Research Fellowship (DGE-1650441). MLMG and IL were students in the Molecular Biology and Genetics Research Experience for Undergraduate (MBG-REU) program, which was supported by DBI1659534. TYC was partially supported by the Einhorn Discovery Grant and Undergraduate Research Fund in the College of Arts and Sciences at Cornell University. We thank the Proteomics and Metabolomics Facility of Cornell University for providing the mass spectrometry data and NIH SIG grant 1S10 OD017992-01 support for the Orbitrap Fusion mass spectrometer. Some strains were obtained from the C. elegans Genetics Center, which is funded by NIH Office of 27 Research Infrastructure Programs (P40 OD010440). The funders had no role in study design, data collection and analysis, decision to publish, or preparation of the manuscript.

**Competing interests:** The authors have declared that no competing interests exist.

**Abbreviations:** BMP, bone morphogenetic protein; BRC, Biotechnology Resource Center; CC, coelomocyte; co-IP, coimmunoprecipitation; EC, extracellular calcium-binding; FS, follistatin-like; GAG, glycosaminoglycan; HS, heparan sulfate; HSPG, heparan sulfate proteoglycan; HSD, honest significant difference; IP, immunoprecipitation; MS, mass spectrometry; POI, protein of interest; RNP, ribonucleoprotein; SDS-PAGE, sodium dodecyl sulfate-polyacrylamide gel electrophoresis; SMOC, secreted modular calcium-binding protein; SP, signal peptide; TGFβ, transforming growth factor β; TY, thyroglobulin type-1; WT, wild type.

Activation of the BMP pathway must be tightly regulated in space, time, level, and duration, as misregulation of the pathway can cause a variety of disorders in humans, including cancer [2,3]. Multiple levels of regulatory mechanisms, including at the extracellular level, have been identified to ensure precise control of BMP signaling [4–6]. One class of extracellular regulators of BMP signaling are the secreted modular calcium-binding proteins (SMOCs).

SMOCs are matricellular proteins belonging to the BM-40/osteonectin/SPARC (basement membrane of 40 kDa/secreted protein acidic and rich in cysteine) family [7]. The SPARC family is characterized as containing an extracellular calcium (EC)-binding domain and a follistatin-like (FS) domain and includes related proteins such as BM-40, SMOCs, QR1, SC1/hevin, tsc36/Flik, and testicans [7]. SMOCs are found in metazoans ranging from flies and nematodes to mice and humans. SMOC homologs characteristically contain a thyroglobulin type-1 (TY) domain followed by an EC domain, and in some cases, contain an FS domain. Notably, the number and arrangement of the domains vary across species. For example, humans have 2 SMOC proteins and both have an FS domain followed by 2 TY domains and 1 EC domain [8,9]. The single *Drosophila* SMOC homolog, Pentagone (Pent), has an FS domain followed by 2 alternating TY and EC domains [10,11], while the lone CeSMOC-1 protein in *Caenorhabditis elegans* has 1 TY domain followed by 1 EC domain [12].

Despite the differences in domain structure arrangements, all SMOC proteins that have been studied regulate BMP signaling [10–16]. The underlying mechanistic bases, however, are not identical across organisms. One of the more unifying models from studies of *Drosophila* Pent and *Xenopus* SMOC-1 suggests that SMOC proteins may function by competing with BMPs for binding to heparan sulfate proteoglycans (HSPGs), thus expanding the range of BMP signaling [17]. However, recent work from *Drosophila* suggests that Pent may have additional unappreciated mechanisms of action in addition to simply competing with Dpp/BMP for binding to HSPGs [18]. For example, Dally/glypican typically functions in a positive fashion as a BMP co-receptor for BMP signaling in *Drosophila* [19–21]. Yet, knocking down enzymes required for the production of HSPGs results in a similar wing disc phenotype as Pent overexpression in the *Drosophila* wing imaginal discs [18], suggesting that the regulatory relationship between these factors is complex. Similarly, *Xenopus* SMOC-1 is known to antagonize BMP signaling by acting downstream of the BMP receptors [16]. Thus, additional studies are needed to further elucidate how SMOC proteins may function to both positively and negatively regulate BMP signaling.

*C. elegans* has a well-conserved BMP-like pathway. The core components of the *C. elegans* BMP pathway include the ligand DBL-1 (BMP), the type I and type II receptors SMA-6 and DAF-4, the R-Smads SMA-2 and SMA-3, and the Co-Smad SMA-4 [22,23]. *C. elegans* also encodes a zinc finger transcription factor SMA-9, a homolog of Schnurri that is known in *Drosophila* and vertebrates to be a co-factor of Smads [24,25]. The BMP pathway is known to regulate multiple biological processes including body size and mesoderm development [22,23,25]. Changes in BMP pathway activation affect body size in a dose-dependent manner with reductions in signaling causing a small (Sma) body size, while increased BMP signaling resulting in a long body length (Lon) phenotype [22]. BMP signaling also regulates postembryonic mesoderm development. In particular, mutations in the zinc finger transcription factor SMA-9 cause a loss of the 2 posterior coelomocytes (CCs) due to a fate transformation in the postembryonic mesoderm or the M-lineage [25]. Paradoxically, mutations in all the known components of the BMP pathway do not exhibit any CC defects on their own, but can suppress the M-lineage defects of *sma-9(0)* mutants (Susm), restoring the proper specification of the 2 postembryonic CCs [25,26]. Both the body size and the Susm phenotypes can be used to assess the functionality of BMP pathway components. In particular, the Susm assay is highly specific and sensitive to altered BMP signaling activity [25,26].

*C. elegans* also has a single SMOC protein, CeSMOC-1, which is known to regulate BMP signaling. We have previously demonstrated that CeSMOC-1 promotes BMP signaling in a cell nonautonomous fashion by acting through the BMP ligand in a positive feedback loop [12]. In this study, we implemented an unbiased approach to identify CeSMOC-1 interacting proteins. We found that, like SMOC proteins in other organisms [10,17,27,28], CeSMOC-1 binds to the HSPG LON-2/glypican via its EC domain. Surprisingly, we also found that full-length CeSMOC-1 can bind to DBL-1/BMP and that CeSMOC-1 can mediate the formation of a LON-2-SMOC-1-DBL-1 tripartite complex in vitro. We showed that while the EC domain of CeSMOC-1 is capable of stimulating BMP signaling when overexpressed, both the TY and the EC domains of CeSMOC-1 are required for full function of CeSMOC-1 when expressed at endogenous levels. We further showed that overexpression of CeSMOC-1 is sufficient to further promote BMP signaling in the absence of LON-2/glypican. Remarkably, site-directed mutations that specifically disrupt CeSMOC-1 interaction with LON-2/glypican resulted in increased BMP signaling as evidenced by mutant worms exhibiting a long body size, while mutations that simultaneously disrupt SMOC-1 interaction with both LON-2/glypican and DBL-1/BMP abolished this effect by rendering the worms non-long. Collectively, our data support a model where a CeSMOC-1-dependent glypican-SMOC-BMP complex inhibits BMP signaling, while the CeSMOC-1-BMP complex promotes BMP signaling.

## Results

### Generation of a *C. elegans* strain with a functionally tagged SMOC-1

To determine how SMOC-1 functions at the molecular level to regulate BMP signaling, we aimed to tag SMOC-1 at the endogenous locus without disrupting its function. We found that endogenously tagging SMOC-1 with GFP at either the N-terminus after the signal peptide (SP) or the C-terminus disrupted SMOC-1 function (S1 Fig). Instead, adding 2 copies of the small FLAG tag to the C-terminus of SMOC-1 at the endogenous locus did not affect SMOC-1 function, based on both the body size and the Susm assays of *smoc-1(jj276[smoc-1::2xflag])* worms (Fig 1). The endogenously tagged SMOC-1::2XFLAG protein, however, was not detectable via western blot (Fig 1B). To overcome this problem, we generated 2 integrated transgenic lines carrying multicopy transgenic arrays that overexpress SMOC-1::2XFLAG, *jjIs5798[smoc-1::2xflag(OE)]* and *jjIs5799[smoc-1::2xflag(OE)]*. Both *jjIs5798* and *jjIs5799* worms exhibit a long phenotype, similar to *jjIs5119[smoc-1(OE)]* worms overexpressing an untagged *smoc-1* (Fig 1C, [12]). These data, combined with the data on *smoc-1(jj276)* worms, suggest that *jjIs5798* and *jjIs5799* worms contain a fully functional *smoc-1::2xflag* that is simply overexpressed. The overexpressed SMOC-1::2xFLAG protein from *jjIs5798* and *jjIs5799* worms is detectable via western blot (Fig 1B). The ability to express and detect a functionally tagged SMOC-1 protein in vivo allowed us to identify SMOC-1-interacting partners from worm extracts and to carry out structure-function studies of the SMOC-1 protein in live worms.

### SMOC-1 associates with LON-2/glypican in worm lysates and when produced in a heterologous system

We next conducted immunoprecipitation (IP) of whole worm lysate from strains overexpressing SMOC-1::2xFLAG and identified co-purifying proteins using mass spectrometry (MS). As controls, we used a strain overexpressing full-length SMOC-1 lacking the FLAG tag (hereafter referred to as "untagged SMOC-1"). We conducted 2 rounds of IP-MS using independent biological samples (see Materials and methods). Analysis of the MS results showed that SMOC-1 was detected exclusively in the tagged samples, indicating that the IP via anti-FLAG

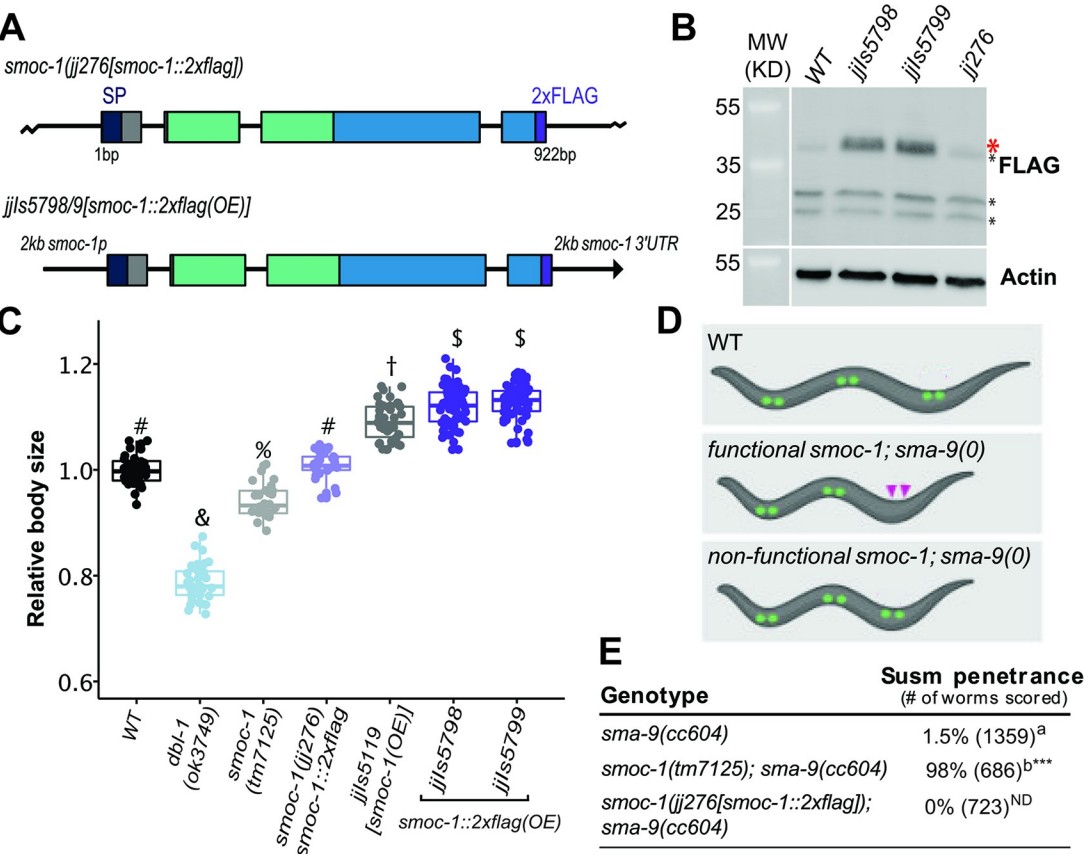

**Fig 1. SMOC-1::2xFLAG is fully functional.** (**A**) Diagrams depicting the *smoc-1::2xflag* endogenous locus (*jj276*) and a transgene with 2 kb *smoc-1* promoter and 2 kb *smoc-1* 3′ UTR flanking the *smoc-1::2xflag* genomic sequence. *jjIs5798* and *jjIs5799* are 2 integrated multicopy array lines that overexpress *smoc-1::2xflag*. In this and all subsequent figures, protein domains in SMOC-1 are indicated by color: SP, signal peptide, navy; TY, thyroglobulin-like domain, green; EC, extracellular calcium binding domain, blue; 2xFLAG, purple. Exons are represented by colored bars, while introns and intergenic regions are represented by thin black lines. (**B**) Western blot of 50 gravid adults of each indicated genotype, probed with anti-FLAG (top) and anti-actin antibodies (bottom). Full-length SMOC-1::2xFLAG at approximately 41 KDa (marked by a red *) is detectable in strains that overexpress SMOC-1::2xFLAG (*jjIs5798* and *jjIs5799*), but not in WT controls nor in *jj276* worms that expresses endogenous SMOC-1::2xFLAG. Black *s indicate nonspecific bands detected by the anti-FLAG antibody. (**C**) Relative body sizes of various strains at the same developmental stage (WT set to 1.0). *jjIs5119* is an integrated multicopy array line that overexpresses untagged *smoc-1*. Groups marked with distinct symbols are significantly different from each other (*P* < 0.001, in all cases when there is a significant difference), while groups with the same symbol are not. Tested using an ANOVA with a Tukey HSD. WT: *N* = 65. *ok3749*: *N* = 38. *tm7125*: *N* = 36. *jj276*: *N* = 31. *jjIs5119*: *N* = 42. *jjIs5798*: *N* = 72. *jjIs5799*: *N* = 68. (**D**) Diagrams depicting the Susm phenotype used to test *smoc-1* functionality. CCs are represented with green circles, with the 2 posterior M-derived CCs marked with pink arrowheads. (**E**) Table showing the penetrance of the Susm phenotype of various mutant strains. The Susm penetrance refers to the percentage of animals with 1 or 2 M-derived CCs as scored using the *arIs37* (*secreted CC::GFP)* reporter (see Materials and methods). For each genotype, the Susm data from 2 independent isolates (see S6 Table) were combined and presented in the table. [a] The lack of M-derived CCs phenotype is not fully penetrant in *sma-9(cc604)* mutants. [b] Data from DeGroot and colleagues [12]. Statistical analysis was conducted by comparing double mutant lines with the *sma-9(cc604)* single mutants. ***P* < 0.001; ND: no difference (unpaired two-tailed Student's *t* test). Original data sets are in S1 Data. CC, coelomocyte; SMOC, secreted modular calcium-binding protein; WT, wild type.

antibody was highly specific to the FLAG tagged SMOC-1. In both experiments, LON-2/glypican was identified as a strong candidate SMOC-1-interaction partner, as demonstrated by the recovery of 13 and 18 respective peptides that map specifically to LON-2 (Fig 2A and S1–S3 Tables). No peptides mapping to LON-2 were detected in the untagged SMOC-1 samples.

To test the interaction between SMOC-1 and LON-2 using an independent assay, we conducted coimmunoprecipitation (co-IP) experiments using *Drosophila* S2 cells that overexpress

**A** >LON-2

MVFRWLILFVLLYRSVLPAEEVVVVDILTNSTSLEEPTEEYTCDCNT
DDLIQKGNYTLTVEVKNMQVRLVEPFR**DAIAFTTGEK**KHLLDFIRFV
TLREVRSTYPLLLNYTDFLNSFDEMIGTFR**NILTTENAISLTGIKYE**
**VSTAVQK**FLSAILPDMFLCLSVGKCRTVPLDYHNCMMASTK**HWSVYL**
**GNTPNKMAMTISEAIYR**YR**KVEFLLVDMHK**QLMNAHNLTLSHECLQE
YVHTLPCNCTMAGITPCHTSCSNSMEKCFGKFSREWAAKLHLMRNMT
STKK**SFLDEFLSLK**KTIFSVIRIFIERK**S̲Y̲V̲Y̲A̲E̲H̲V̲F̲NSCGPLGEMI**
**IHPSKHSVHFQSPGPFVS**RGD**GAVRELQLSAKTWD**RLGR**KICDHSGV**
**VLNPTMCYDGTKVISIDHDLLPIT**KDVRPRPMMDWIEKKNEKKANVE
GSASNPLWDDEDSEDFDGSGSGMPPVIDRNPVKAIIQEDHPK**NIDLS**
**TNPKGPSVIVTEK**EIQPDGSTTSSILICVIIVAVIKLF

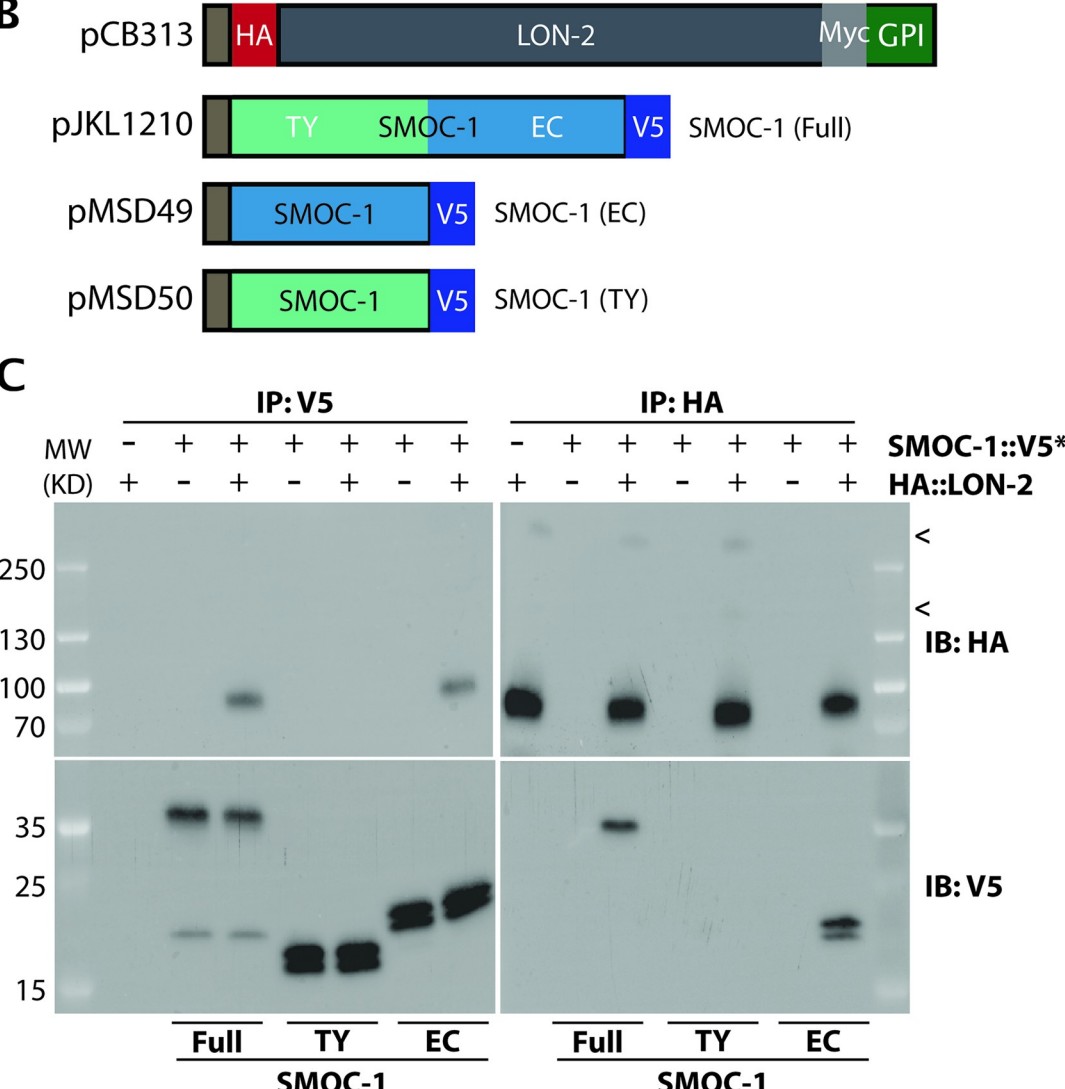

**B**

pCB313 | HA | LON-2 | Myc | GPI

pJKL1210 | TY | SMOC-1 | EC | V5 | SMOC-1 (Full)

pMSD49 | SMOC-1 | V5 | SMOC-1 (EC)

pMSD50 | SMOC-1 | V5 | SMOC-1 (TY)

**C**

IP: V5    IP: HA

MW
(KD)

IB: HA

IB: V5

Full  TY  EC         Full  TY  EC
SMOC-1              SMOC-1

**Fig 2. SMOC-1 interacts with LON-2 in worm extracts and in vitro.** (**A**) Sequence of *C. elegans* LON-2 protein with peptide regions detected in the different IP/MS experiments marked. Peptides detected in the full-length SMOC-1::2xFLAG pulldown are underlined, while peptides detected in the SMOC-1(EC)::2xFLAG pulldown are in bold. Important regions of LON-2 are also highlighted, including the signal peptide (gray), RGD motif (orange), RLGR consensus putative furin protease recognition site (green), HS GAG attachment sites (aqua), and GPI linkage site (red). The 3 residues (S311, A315, and F319) in red boxes

are predicted to mediate LON-2-SMOC-1 interaction and are mutated to generate the *lon-2(mut)* mutation described later. (**B**) Diagrams of the expression constructs used in the *Drosophila* S2 cell expression system. (**C**) Results of co-IP experiments testing the interaction of HA::LON-2 with different versions of SMOC-1::V5* produced in *Drosophila* S2 cells, including SMOC-1(Full)::V5, SMOC-1(TY)::V5, and SMOC-1(EC)::V5. IP with anti-V5 beads or anti-HA beads, IB with anti-HA or anti-V5 antibodies, as indicated. Experiments were independently repeated in triplicate, with representative results shown in this figure, and < points to faint bands that may represent glycosylated LON-2. We do not know whether posttranslational modification or cleavage causes SMOC-1 proteins to run as 2 bands when expressed in *Drosophila* S2 cells. Original images of western blots can be found in S1 Raw Images. co-IP, coimmunoprecipitation; EC, extracellular calcium-binding; GAG, glycosaminoglycan; HS, heparan sulfate; IB, immunoblot; IP, immunoprecipitation; SMOC, secreted modular calcium-binding protein; TY, thyroglobulin type-1.

HA::LON-2 and SMOC-1::V5 (Fig 2B). As shown in Fig 2C, we detected association between SMOC-1 and LON-2 in bidirectional IP experiments: IP of SMOC-1::V5 pulled down HA::LON-2, while IP of HA::LON-2 pulled down SMOC-1::V5.

Taken together, results from our co-IP experiments using both worm extracts and the *Drosophila* S2 cell expression system demonstrated that SMOC-1 interacts with the glypican LON-2.

## SMOC-1 interacts with LON-2/glypican through its EC domain

To identify the specific domain via which SMOC-1 interacts with LON-2, we expressed tagged versions of the SMOC-1 EC domain or the SMOC-1 TY domain in *Drosophila* S2 cells, and tested their interaction with LON-2. Bi-directional co-IP experiments showed that SMOC-1(EC), but not SMOC-1(TY), can interact with LON-2 (Fig 2C). To corroborate these co-IP results, we generated transgenic lines overexpressing SMOC-1(EC)::2xFLAG and conducted IP-MS experiments using the same condition used for IP-MS experiments with full-length SMOC-1::2xFLAG (see Materials and methods). Results from these experiments supported an interaction between SMOC-1(EC) and LON-2, as 15 peptides that map specifically to LON-2 were recovered (Fig 2A and S1 Table). The co-IP experiments using both worm extracts and the *Drosophila* S2 cells demonstrated that the EC domain of SMOC-1 interacts with LON-2/glypican.

## The EC domain of SMOC-1 is sufficient to promote BMP signaling when overexpressed

We then tested the functionality of the SMOC-1 TY and EC domain individually by generating transgenic lines overexpressing either the TY domain or the EC domain tagged with 2xFLAG (Fig 3A). We found that when overexpressed, SMOC-1(EC) rescued the body size defect of *smoc-1(0)* animals, making the worms longer than wild-type (WT) worms, although not as long as worms overexpressing the fully functional full-length SMOC-1 (Figs 1 and 3C). Overexpressed SMOC-1(EC) also robustly rescued the Susm phenotypes of *smoc-1(0)* worms (Fig 3D). In contrast, overexpression of SMOC-1(TY) failed to rescue the *smoc-1(0)* body size (Fig 3C), and only slightly rescued the Susm phenotype of *smoc-1(0)* mutants (Fig 3D). The lack of rescue by the SMOC-1(TY) domain is not due to failed expression of SMOC-1(TY). As shown in Fig 3B, both SMOC-1(EC)::2xFLAG and SMOC-1(TY)::2xFLAG were detectable on western blots in worms overexpressing each corresponding domain. Taken together, our results indicate that the EC domain of SMOC-1 is both necessary and sufficient to regulate BMP signaling when overexpressed.

## The SMOC-1 EC domain is not fully functional when expressed at the *smoc-1* endogenous locus

To determine if SMOC-1(EC) is sufficient to function in BMP signaling when expressed at the endogenous level, we used CRISPR to alter the *smoc-1* genomic locus and generated an allele

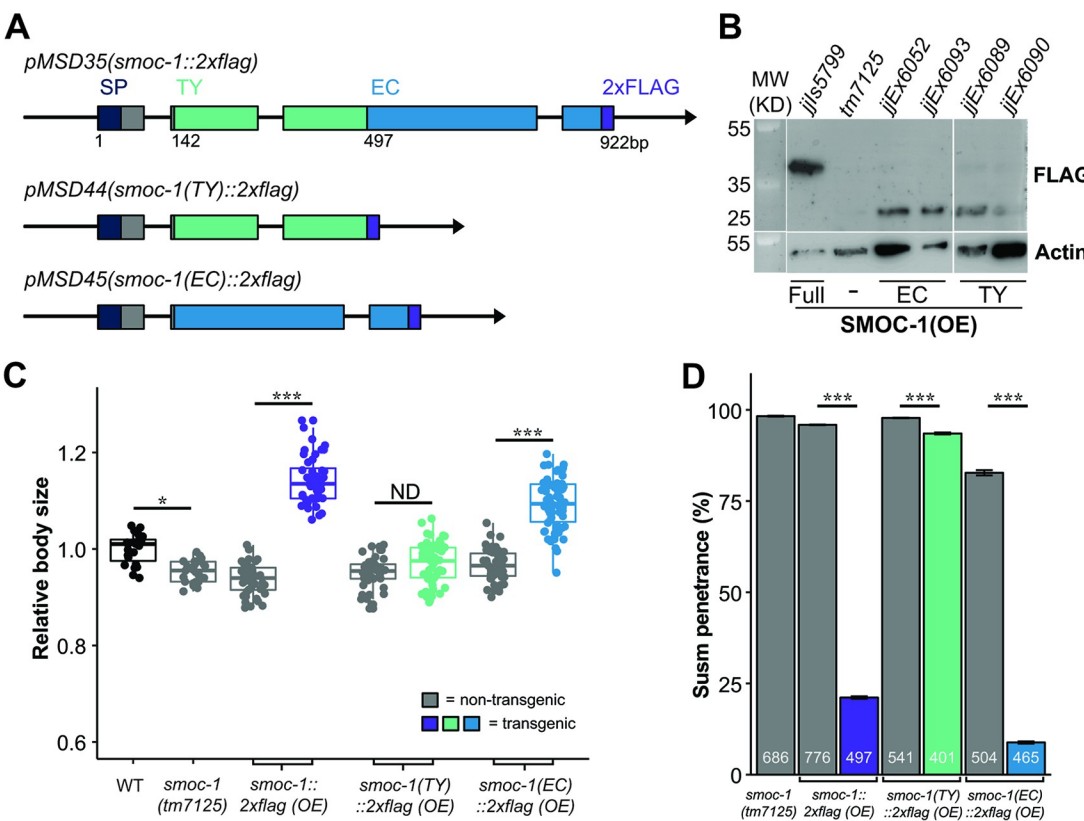

**Fig 3. The EC domain of SMOC-1 is sufficient to regulate BMP signaling when overexpressed.** (**A**) Diagrams depicting the genomic constructs expressing full-length SMOC-1::2xFLAG (pMSD35), SMOC-1(TY)::2xFLAG (pMSD44), and SMOC-1 (EC)::2xFLAG (pMSD45). All plasmids contain the same 2 kb promoter and 2 kb 3′ UTR of *smoc-1*. SMOC-1(TY) ends at amino acid 134, while SMOC-1(EC) starts at amino acid 135, both containing the same SMOC-1 SP, and the 2xFLAG tag. (**B**) Western blot of 50 gravid adults of each indicated genotype, probed with anti-FLAG (top) and anti-actin antibodies (bottom). The strain overexpressing full-length SMOC-1::2xFLAG (*jjIs5799*) is an integrated transgenic strain, while those overexpressing SMOC-1 (TY)::2xFLAG (*jjEx6089* and *jjEx6090*) or SMOC-1(EC)::2xFLAG (*jjEx6052* and *jjEx6093*) carry the transgenes as extra chromosomal arrays, thus the expression level appeared lower due to random loss of the array during each mitotic division. (**C**) Relative body sizes of strains carrying indicated versions of *smoc-1* as extra chromosomal arrays in a *smoc-1(tm7125)* null background at the same developmental stage (WT set to 1.0). For panels C and D, gray indicates non-transgenic worms that do not express any *smoc-1*. Two independent transgenic lines were measured and combined for each plasmid being tested here. Statistical analysis was done by comparing transgenic strains with non-transgenic counterparts. ***$P < 0.001$; *$P < 0.01$, ND: no difference (ANOVA followed by Tukey HSD). WT: $N = 23$. *tm7125*: $N = 25$. Full-length *smoc-1* (transgenic: $N = 44$; non-transgenic: $N = 41$). *smoc-1(TY)* (transgenic: $N = 40$; non-transgenic: $N = 39$). *smoc-1(EC)* (transgenic: $N = 66$; non-transgenic: $N = 47$). (**D**) Summary of the Susm penetrance of strains carrying indicated versions of *smoc-1* in a *smoc-1(tm7125); sma-9 (cc604)* background. The Susm penetrance refers to the percent of animals with 1 or 2 M-derived CCs as scored using the *arIs37 (secreted CC::GFP)* reporter. For each genotype, 2 independent isolates were generated (as shown in the strain list), the Susm data from the 2 isolates were combined and presented here. Number of animals scored are noted on each bar. Statistical analysis was done to compare transgenic strains with non-transgenic counterparts. See panel C for color legend. ***$P < 0.001$ (general linear model, Wald statistic). Original data sets are in S1 Data. Original images of western blots can be found in S1 Raw Images. BMP, bone morphogenetic protein; EC, extracellular calcium-binding; SMOC, secreted modular calcium-binding protein; SP, signal peptide; TY, thyroglobulin type-1; WT, wild type.

(*jj441)* that expresses SMOC-1(EC) and 2 alleles (*jj411* and *jj412*) that express SMOC-1(TY) (Fig 4A). We then tested the functionality of SMOC-1(EC) and SMOC-1(TY) using both the body size assay and the more sensitive Susm assay. We found that while SMOC-1(EC)-expressing worms do not exhibit any body size defect, SMOC-1(TY)-expressing worms are even smaller than *smoc-1(0)* null mutant worms (Fig 4B). Furthermore, both SMOC-1(EC)-expressing worms and SMOC-1(TY)-expressing worms exhibited a partially penetrant Susm phenotype (Fig 4C). Since worms carrying the WT *smoc-1* locus do not display any Susm

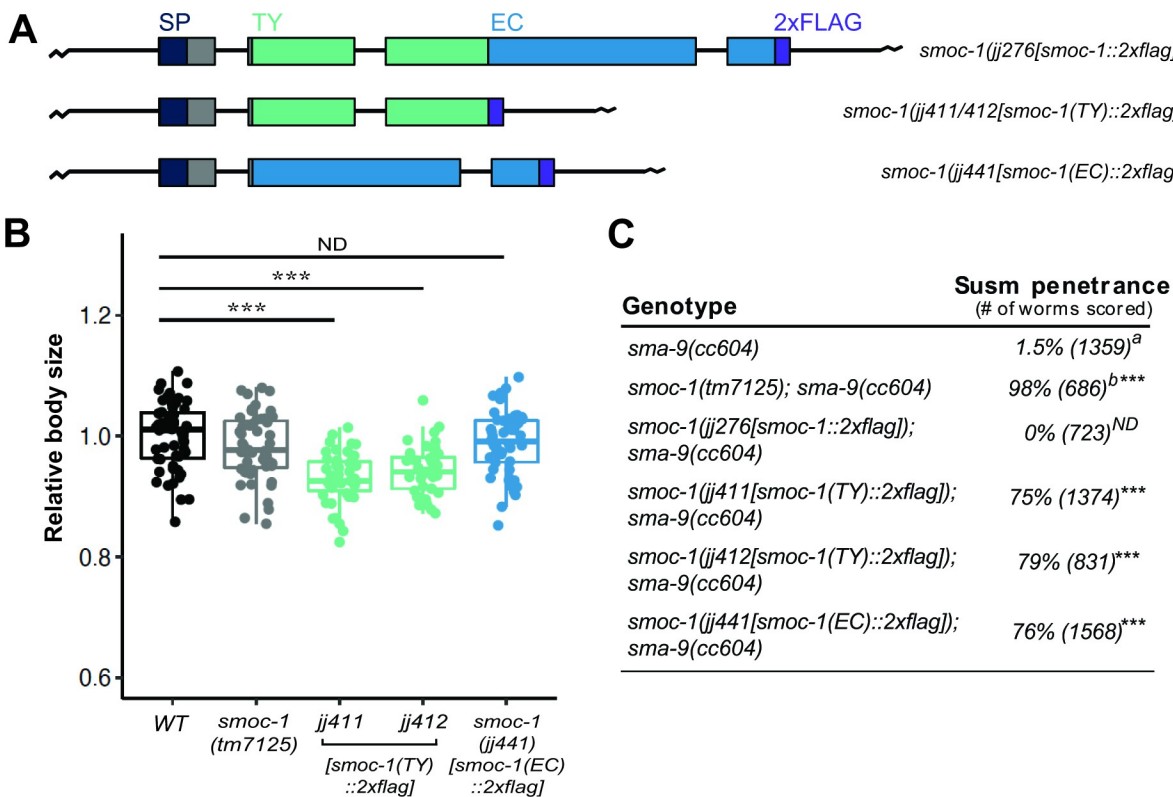

**Fig 4. When expressed at the endogenous locus, neither SMOC-1(TY) nor SMOC-1(EC) is fully functional in regulating BMP signaling.** (**A**) Diagrams depicting the full-length *smoc-1*, as well as truncated *smoc-1(TY)* and *smoc-1(EC)* at the endogenous locus. All of them are tagged with a 2xFLAG tag at the C-terminal end. Color keys are the same as in Fig 1A. (**B**) Relative body sizes of worms at the same developmental stage WT (WT set to 1.0). ***$P < 0.001$; ND: no difference (ANOVA followed by Tukey HSD). WT: $N = 54$. *tm7125*: $N = 51$. *jj441*: $N = 51$. *jj411*: $N = 47$. *jj412*: $N = 41$. (**C**) Table showing the penetrance of the Susm phenotype of *smoc-1(TY)* and *smoc-1(EC)* as compared to the *smoc-1(0)* worms. [a] The lack of M-derived CCs phenotype is not fully penetrant in *sma-9(cc604)* mutants. [b] Data from DeGroot and colleagues [12]. Statistical analysis was conducted by comparing double mutant lines with the *sma-9(cc604)* single mutants. ***$P < 0.001$; ND: no difference (unpaired two-tailed Student's *t* test). Original data sets are in S1 Data. CC, coelomocyte; EC, extracellular calcium-binding; SMOC, secreted modular calcium-binding protein; TY, thyroglobulin type-1; WT, wild type.

phenotype (Fig 4C), our results suggest that both the TY and the EC domains are together required for full function of SMOC-1 when expressed at the endogenous level.

## The BMP ligand DBL-1 binds to full-length SMOC-1, but not LON-2/glypican, in vitro

Previous studies of the *Drosophila* and *Xenopus* SMOC proteins led to a model where SMOCs function by competing with BMP ligands for binding to HSPGs, allowing the spreading of BMP ligands [17]. We have found that SMOC-1 can associate with LON-2/glypican via its EC domain. We therefore decided to further investigate the relationship between SMOC-1, the BMP ligand DBL-1, and LON-2/glypican, by expressing differently tagged SMOC-1, DBL-1 and LON-2 proteins using the S2 cell expression system (Fig 5). Since BMP molecules are produced as inactive molecules with a prodomain attached to the mature active domain ([29]), we generated constructs that would allow us to detect not only the full-length proteins, but also both the prodomain and the mature (active) domain of DBL-1 after processing and secretion into the media (Fig 5A and 5C). We then performed reciprocal co-IP experiments for each protein pair. We did not detect any association between LON-2/glypican with either full-

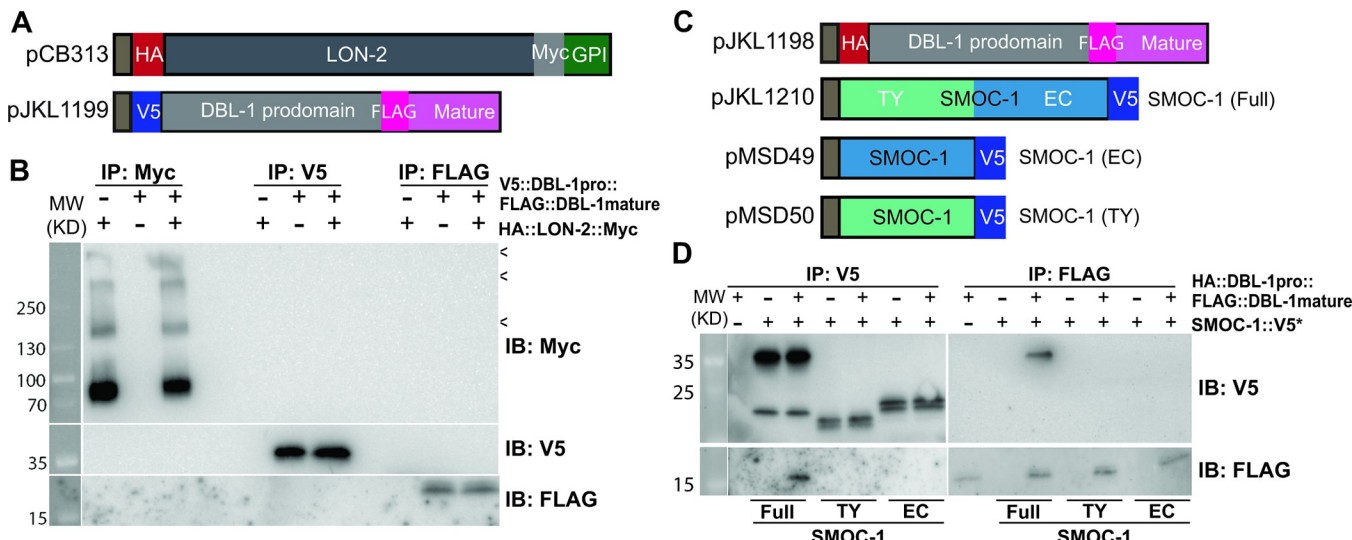

**Fig 5. SMOC-1, but not LON-2, binds to DBL-1 when expressed in S2 cells.** (**A**) Diagrams of LON-2 and DBL-1 expression constructs used in the *Drosophila* S2 cell expression system. (**B**) Results of co-IP experiments testing the interaction between LON-2::Myc and V5::DBL-1 prodomain::FLAG::DBL-1 mature domain. IP with anti-Myc beads, anti-V5 beads or anti-FLAG beads and IB with anti-Myc, anti-V5, or anti-FLAG antibodies, as indicated. Experiments were independently repeated in triplicate, with representative results shown in this figure, and < points to faint bands that likely represent glycosylated LON-2. (**C**) Diagrams of SMOC-1 and DBL-1 expression constructs used in the *Drosophila* S2 cell expression system. (**D**) Results of co-IP experiments testing the interaction between HA::DBL-1 prodomain::FLAG::DBL-1 mature domain and different versions of SMOC-1::V5*. IP with anti-V5 beads or anti-FLAG beads and IB with anti-V5 or anti-FLAG antibodies, as indicated. The source of DBL-1 in these experiments was cell media, which does not contain full-length DBL-1, but only HA-tagged prodomain and FLAG-tagged mature domain. Experiments were independently repeated in triplicate, with representative results shown in this figure. Original images of western blots can be found in S1 Raw Images. co-IP, coimmunoprecipitation; EC, extracellular calcium-binding; IB, immunoblot; IP, immunoprecipitation; SMOC, secreted modular calcium-binding protein; TY, thyroglobulin type-1.

length DBL-1 (S2 Fig) or each of the domains of DBL-1 (Fig 5B). Instead, full-length SMOC-1, but neither SMOC-1(EC) nor SMOC-1(TY), can co-immunoprecipitate with the mature domain of DBL-1 (Fig 5D).

## In silico structural modeling supports the interaction between LON-2/glypican and SMOC-1 and between SMOC-1 and DBL-1/BMP

The interaction between SMOC-1 and the mature domain of DBL-1/BMP was unexpected, given previous studies of SMOC proteins in other systems. We therefore sought independent ways to verify this finding. Recent advances in protein structure prediction have enabled the structures of protein–protein interactions to be modeled in silico ([30–33]). These predicted structural models can serve as useful tools for interpreting functional results and for generating hypotheses that can be tested through further experimentation. We used the ColabFold [34] implementation of AlphaFold2 [30] to determine whether confident structural predictions could be generated for the interactions between LON-2, SMOC-1, and mature DBL-1. Consistent with results of our physical interaction experiments, this structural modeling predicted a strong interaction between LON-2 and SMOC-1, and their interaction involves the EC domain of SMOC-1 (Fig 6A). None of the possible forms of DBL-1 (mature domain, prodomain, or full-length) were predicted to interact with LON-2 (S3 Fig). However, the mature domain of DBL-1 and SMOC-1 were predicted to interact, and the predicted interaction involves a bipartite interaction with both the TY domain and the C-terminal portion of the EC domain of the full-length SMOC-1 protein (Figs 6B and S3). All these predictions are consistent with results of our physical interaction experiments.

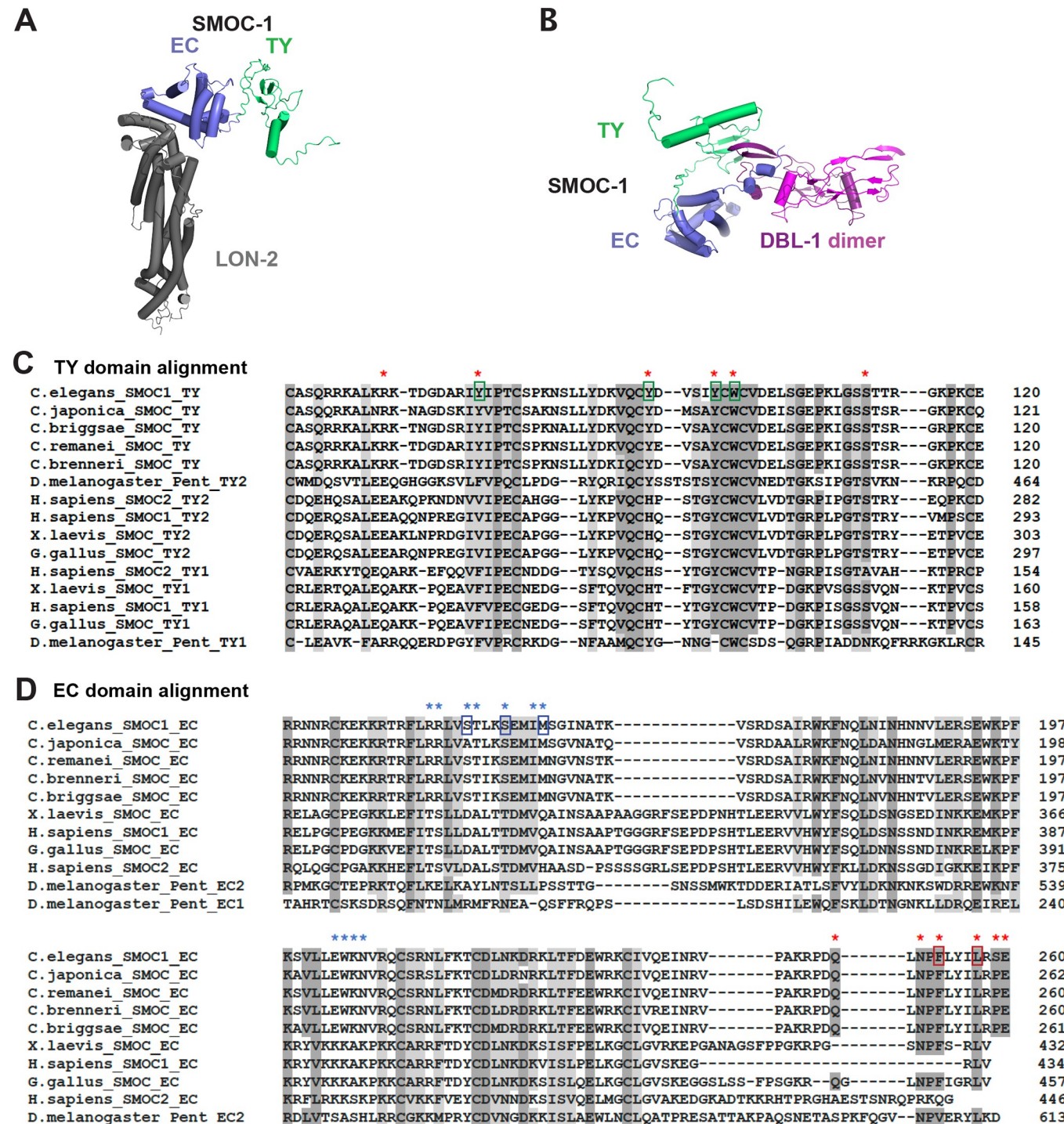

**Fig 6. Structural modeling of interactions between SMOC-1 and LON-2, and between SMOC-1 and DBL-1.** (**A**) Predicted structure of a complex formed between LON-2 and SMOC-1. The EC domain of SMOC-1 is predicted to interact with LON-2. (**B**) Predicted structure of a complex formed between SMOC-1 and a homodimer of the DBL-1 mature domain. Both the EC and TY domains of SMOC-1 are predicted to interact with DBL-1. (**C**) Multiple sequence alignment of the TY domains of SMOC-1 homologs using Clustal Omega (CLUSTAL O(1.2.4)). Red * marks the residues at the interface between SMOC-1 and DBL-1, as identified via ColabFold. Green colored boxes indicate residues (Y72, Y90, Y95, W97) mutated to generate the SMOC-1(M3) mutation. (**D**) Multiple sequence alignment of the EC domains of SMOC-1 homologs using Clustal Omega (CLUSTAL O(1.2.4)). Red * marks the residues at the interface between SMOC-1 and DBL-1, and blue * marks the residues at the interface between SMOC-1 and LON-2, as identified via ColabFold. In both C and D, dark shaded residues are identical, while light shaded residues are conserved, among all or most of the homologs. Blue colored boxes indicate residues (S152, S156,

M160) mutated to generate the SMOC-1(M1) mutation, while red boxes indicate residues (F253, L257) mutated to generate the SMOC-1(M2) mutation. EC, extracellular calcium-binding; SMOC, secreted modular calcium-binding protein; TY, thyroglobulin type-1.

Importantly, our structural modeling also identified key residues located at the interfaces between each pair of interacting partners. Sequence comparisons between the *C. elegans* SMOC-1, LON-2, DBL-1, and their corresponding counterparts in other organisms, ranging from other nematode species to *Drosophila* and mammals, showed that these key residues are highly conserved among each of the 3 protein families (Figs 6C and 6D, S4 and S5). In particular, an NPF/VxxxL motif at the C-terminal end of the EC domain, which is predicted to interact with DBL-1/BMP, appears to be conserved among SMOC homologs in *Drosophila*, *Xenopus*, and chicken.

## Predicted interactions between SMOC and BMP2/4 ligand appear conserved in other eukaryotes

To investigate whether the interactions we observed between CeSMOC-1 and DBL-1/BMP might be conserved in other organisms, we performed additional structural predictions using vertebrate and *Drosophila* SMOC proteins and the respective vertebrate BMP2/4 or *Drosophila* DPP (see Materials and methods). Remarkably, strong interactions were predicted between SMOC-1 and DBL-1 homologs in *H. sapiens*, *X. laevis*, and *D. melanogaster* (Fig 7). The predicted interactions share similarities with the predictions for *C. elegans* SMOC-1 and DBL-1 described above. Each of these other SMOC proteins is predicted to interact with its corresponding mature domain of the BMP ligand via a TY domain interaction that is similar to that observed in the *C. elegans* prediction (Fig 6). But there are also interesting differences that appear to reflect the different domain organization of SMOC proteins in these other organisms. The SMOC proteins from *H. sapiens*, *X. laevis*, and *D. melanogaster* each contain 2 TY domains, and in each case both TY domains are predicted to make a similar contact with each of the 2 BMP molecules in the BMP dimer. Furthermore, while the interaction between the C-terminal portion of the EC domain and BMP is found in the predictions of the *C. elegans*, *X. laevis*, and *D. melanogaster* SMOC-BMP complexes, it is not present in the *H. sapiens* prediction. However, *H. sapiens* SMOC1 appears to replace the EC domain interaction with a portion of the SMOC1 polypeptide located in between the 2 TY domains. This additional interaction involving the linker in between TY domains is also found in the prediction of the *X. laevis* complex. Therefore, these predictions support the notion that the interaction between SMOC and BMP is conserved in both invertebrate and vertebrate organisms.

## LON-2/glypican, SMOC-1, and DBL-1/BMP forms a tripartite complex in vitro

Our structural modeling further predicted the presence of a SMOC-1-dependent tripartite complex between LON-2/glypican, SMOC-1, and DBL-1/BMP, in which LON-2 interacts with SMOC-1 and SMOC-1 interacts with mature DBL-1 (Fig 8A and 8B). We experimentally tested this in silico structural prediction, using proteins expressed in the *Drosophila* S2 expression system. We performed co-IP experiments by mixing LON-2 and DBL-1 together in the presence or absence of SMOC-1. Again, the mature domain of DBL-1 can co-IP SMOC-1, and this interaction is irrespective of whether LON-2 is present or not (Fig 8C). In the same co-IP experiments, immunoprecipitating the mature domain of DBL-1 pulled down LON-2 only in the presence of SMOC-1 (Fig 8C). These results strongly suggest that SMOC-1 can mediate

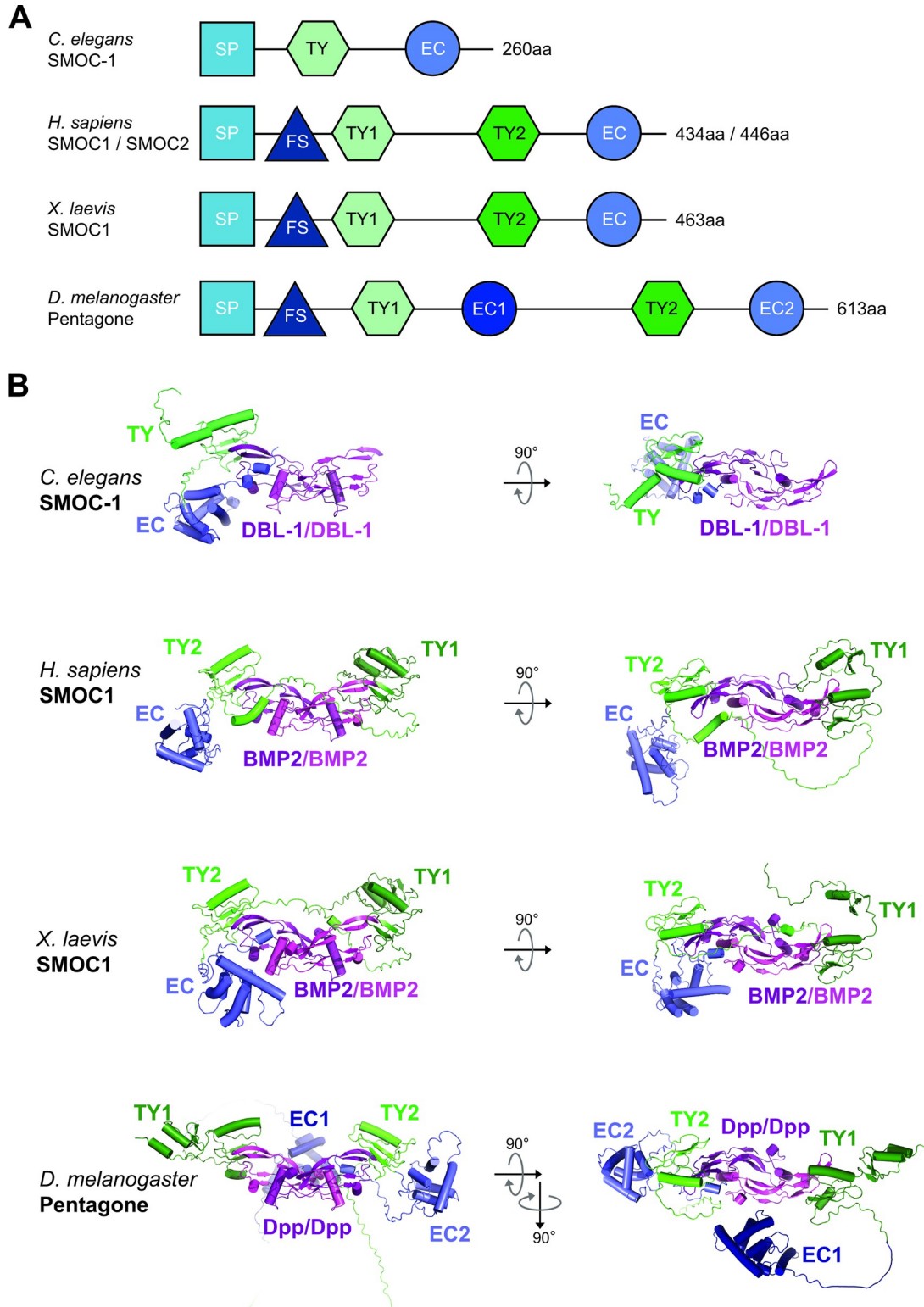

**Fig 7. Structural modeling of interactions between SMOC and BMP homologs from both vertebrate and invertebrate species.** (**A**) Diagrams of SMOC proteins from *C. elegans*, *H. sapiens*, *X. laevis*, and *D. melanogaster*, showing the unique arrangement of the TY and EC domains in each SMOC protein. (**B**) Predicted structures of complexes formed between homologs of CeSMOC-1 and mature DBL-1 from *C. elegans*, *H. sapiens*, *X. laevis*, and *D. melanogaster*. Predictions are similar between HsSMOC1 and HsSMOC2, but only HsSMOC1 is shown here. Predictions are also similar between BMP4 and BMP2

for *H. sapiens* and *X. laevis* SMOCs, but only BMP2 is shown here. Note that each of the BMP ligands is predicted to make a similar interaction with a TY domain of its corresponding SMOC-1 homolog. The interaction with the C-terminus of the EC domain appears to be somewhat less conserved, as it is present in *X. laevis* and *D. melanogaster* predictions, but not in the *H. sapiens* predictions. However, *H. sapiens* SMOC1 appears to replace the EC domain interaction with a portion of the SMOC1 polypeptide located in between the 2 TY domains. This additional interaction involving the linker in between TY domains is also found in the prediction of the *X. laevis* complex. The SMOC-1 homologs from *H. sapiens*, *X. laevis*, and *D. melanogaster* each contain 2 TY domains, and in each prediction both domains make a similar contact with each of the 2 BMP molecules in the BMP dimer. BMP, bone morphogenetic protein; EC, extracellular calcium-binding; FS, follistatin-like; SMOC, secreted modular calcium-binding protein; TY, thyroglobulin type-1.

LON-2 and DBL-1 interaction and that the 3 proteins can form a SMOC-1-dependent tripartite complex.

## Mutations specifically disrupting SMOC-1-LON-2 interaction result in a long body size phenotype

We next sought in vivo evidence to determine the functional significance of SMOC-1-mediated interactions with LON-2 and DBL-1. Our structural models identified key residues in both SMOC-1 and LON-2 that are predicted to mediate their interaction, S152 S156 M160 in SMOC-1, and S311 A315 F319 in LON-2. We first generated a mutant form of SMOC-1 in the putative LON-2-binding domain, which we named SMOC-1(M1) that carries 3 mutations, S152D S156D M160D, in the EC domain (Fig 9A). In co-IP assays using proteins produced in *Drosophila* S2 cells, SMOC-1(M1) specifically failed to interact with LON-2, but retained interaction with DBL-1 (Fig 9B and 9C). We then introduced the same amino acid substitutions in the endogenous *smoc-1::2xflag* locus [*smoc-1(jj276)*] via CRISPR and obtained 2 alleles, *jj499* and *jj500*. Remarkably, both *jj499* and *jj500* mutant worms are long (Fig 9D and 9E). Furthermore, both *jj499* and *jj500* exhibit partial dominance, as heterozygous animals carrying the mutation are longer than WT animals (Fig 9E).

We then generated a mutant form of LON-2, LON-2(mut), that is predicted to disrupt its interaction with SMOC-1, S311D A315D F319D (Fig 10A). Again, LON-2(mut) protein produced in *Drosophila* S2 cells failed to interact with SMOC-1 in co-IP assays (Fig 10B). When the same mutations were introduced into the endogenous *lon-2* locus via CRISPR, the resulting mutant worms (2 alleles, *jj507* and *jj508*) were long and as long as the *lon-2(e678)* null worms (Fig 10C and 10D).

Taken together, these results suggest that SMOC-1 and LON-2 interaction has a negative impact on BMP signaling and that SMOC-1 proteins that do not interact with LON-2 can positively promote BMP signaling.

## Mutations in SMOC-1 that disrupt its interaction with both LON-2 and DBL-1 do not cause a long body size phenotype

We reasoned that the ability of SMOC-1(M1) to promote BMP signaling is due to its interaction with DBL-1. To test this hypothesis, we sought to disrupt the interaction between SMOC-1 and DBL-1. Our structural models identified key residues in both the EC domain and the TY domain of SMOC-1 that are predicted to mediate SMOC-1-DBL-1 interaction, F253 L257 in the EC domain and Y72 Y90 Y95 W97 in the TY domain (Fig 9A). We generated 2 mutant forms of SMOC-1, SMOC-1(M2) that carries the mutations F253D L257D, and SMOC-1(M3) that carries the mutations Y72A Y90A Y95A W97A (Fig 9A). In co-IP assays using proteins produced in *Drosophila* S2 cells, both SMOC-1(M2) and SMOC-1(M3) could still bind LON-2, but exhibited a significant decrease in their ability to bind DBL-1 (Fig 9B and 9C). We then attempted to introduce the same M2 or M3 mutations in the SMOC-1(M1) background via

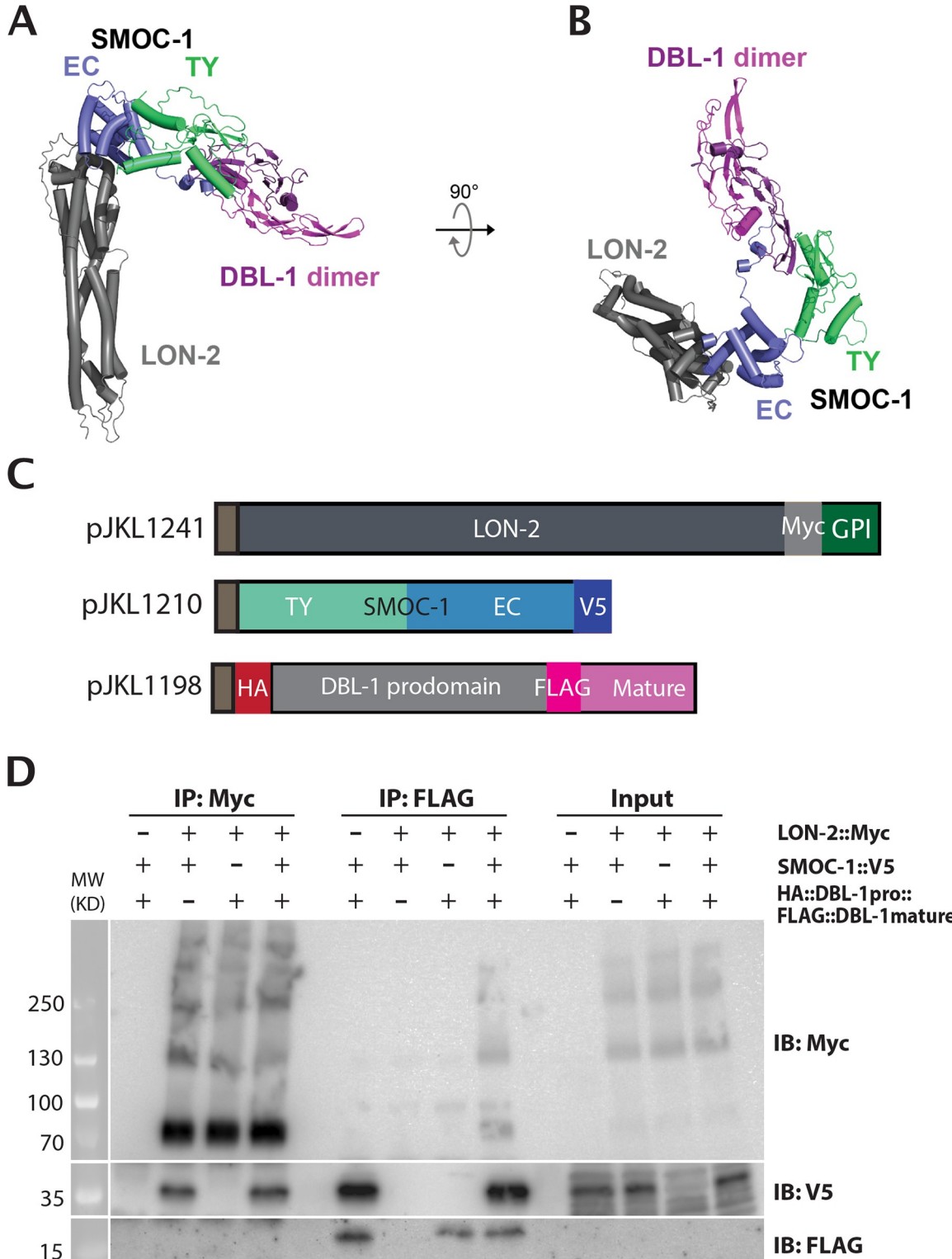

**Fig 8. Co-IP results testing the model for tripartite complex formation.** (**A**) Predicted structure of a complex formed between LON-2, SMOC-1, and a homodimer of the DBL-1 mature domain. The membrane-anchoring region of LON-2 lies at the bottom of the panel. (**B**) The same structure prediction as in A, but shown from the "top." (**C**) Diagrams of LON-2, SMOC-1, and DBL-1 expression constructs used in the *Drosophila* S2 cell expression system. (**D**) Results of co-IP experiments testing the interaction between LON-2::Myc and HA::DBL-1 prodomain::FLAG::DBL-1 mature domain in the presence or absence of SMOC-1::V5. IP with anti-FLAG beads,

or anti-Myc beads, and IB with anti-Myc, anti-V5 or anti-FLAG antibodies, as indicated. Note that immunoprecipitation of FLAG::DBL-1mature domain with anti-FLAG beads can pull down LON-2::Myc only in the presence of SMOC-1::V5. No FLAG::DBL-1mature protein was detected when IP was performed using anti-Myc antibodies, very likely due to the low amount of FLAG::DBL-1mature protein in the co-IP experiments, as FLAG::DBL-1mature protein was not detectable in the Input lanes where approximately 1% of materials used for the co-IP experiments were loaded. Experiments were independently repeated in triplicate, with representative results shown in this figure. Original images of western blots can be found in S1 Raw Images. co-IP, coimmunoprecipitation; EC, extracellular calcium-binding; IB, immunoblot; IP, immunoprecipitation; SMOC, secreted modular calcium-binding protein; TY, thyroglobulin type-1.

CRISPR. We failed to recover worms carrying the M3 mutation, but obtained 2 alleles, *jj499 jj510* and *jj499 jj511*, that carry both the M1 and M2 mutations [SMOC-1(M1+M2)]. As shown in Fig 9F, both SMOC-1(M1+M2) mutant worms are not long, as was seen with SMOC-1(M1) mutants (Fig 9D and 9F). To rule out the possibility that the body size phenotype is due to the mutant SMOC-1 proteins being unstable, we generated transgenic lines that overexpress SMOC-1(M1), SMOC-1(M2), or SMOC-1(M3) in the *smoc-1* null background. All mutant proteins from the transgenic animals are detectable on western blots (S6 Fig). Collectively, these results suggest that the ability of SMOC-1(M1) to promote BMP signaling is dependent on its interaction with DBL-1.

### Additional evidence supporting a LON-2-independent role of SMOC-1 in promoting BMP signaling

Our results so far support a negative, LON-2-dependent role and a positive, DBL-1-dependent, yet LON-2-independent role of SMOC-1 in regulating BMP signaling. We sought additional evidence to test this hypothesis. We generated strains that overexpress *smoc-1 (jjIs5799[smoc-1 (OE)])*, or *dbl-1 (jjIs6448[dbl-1(OE)])* (S7 Fig) and measured the body sizes of animals overexpressing *smoc-1* in the presence or absence of WT *lon-2*, and in the presence or absence of *dbl-1 (OE)*. Based on our hypothesis, we predicted that drastically overexpressing SMOC-1 can promote BMP signaling irrespective of the presence or absence of LON-2. Moreover, overexpressing SMOC-1 would be expected to further augment the positive effect of increased levels of DBL-1/BMP on BMP signaling. As shown in Fig 11, *smoc-1(OE)* animals and *dbl-1(OE)* animals are both significantly longer than *lon-2(e678)* null animals. Notably, *smoc-1(OE); lon-2(e678)* double mutant animals are similar in length to *smoc-1(OE)* animals yet longer than *lon-2(e678)* null animals, demonstrating that overexpression of SMOC-1 can promote BMP signaling in the absence of LON-2. Similar results were observed between *dbl-1(OE)* as compared to *dbl-1(OE); lon-2(e678)* animals. Remarkably, double mutants that overexpress both SMOC-1 and DBL-1 are longer than either *smoc-1(OE)* or *dbl-1(OE)* single mutants (Fig 11). These genetic findings are consistent with the dual functions of SMOC-1 in regulating BMP signaling.

### Discussion

In this study, we discovered that *C. elegans* SMOC-1 binds the glypican LON-2, as well as the mature domain of the BMP protein DBL-1. Moreover, CeSMOC-1 can mediate the formation of a glypican-SMOC-BMP tripartite complex. Our biochemical and molecular genetic data together support a model for dual functionality of CeSMOC-1, acting both positively and negatively in the BMP pathway.

### A model for how CeSMOC-1 functions to regulate BMP signaling

LON-2/glypican is well established to be a negative regulator of BMP signaling in *C. elegans* [35]. We have previously shown that the other glypican in the *C. elegans* genome, GPN-1/

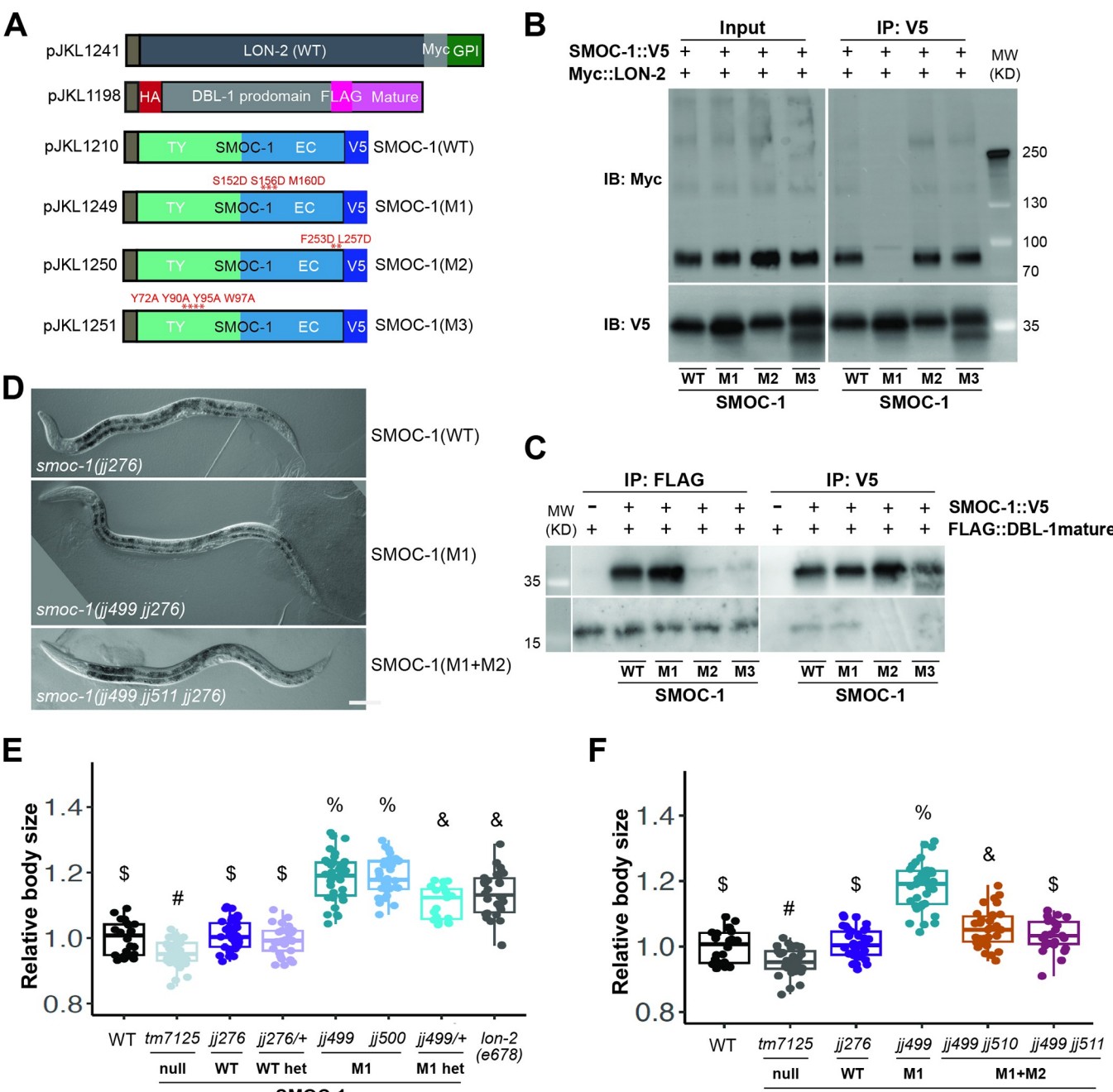

**Fig 9. The in vitro and in vivo consequences of mutations in SMOC-1 that affect either LON-2 binding or DBL-1 binding.** (**A**) Diagrams of LON-2, DBL-1, WT, and mutant SMOC-1 expression constructs used in the *Drosophila* S2 cell expression system. (**B**) Results of co-IP experiments testing the interaction between LON-2::Myc and V5-tagged WT and mutant SMOC-1 proteins. IP with anti-V5 beads and IB with anti-Myc or anti-V5 antibodies, as indicated. Experiments were independently repeated in triplicate, with representative results shown in this figure. (**C**) Results of co-IP experiments testing the interaction between HA::DBL-1 prodomain::FLAG::DBL-1 mature domain and V5-tagged WT and mutant SMOC-1 proteins. IP with anti-V5 beads or anti-FLAG beads and IB with anti-V5 or anti-FLAG antibodies, as indicated. The source of DBL-1 in these experiments was cell media, which does not contain full-length DBL-1, but only HA-tagged prodomain and FLAG-tagged mature domain. Experiments were independently repeated in triplicate, with representative results shown in this figure. (**D**) DIC images of worms endogenously expressing SMOC-1(WT) *[smoc-1(jj276)]*, SMOC-1(M1) *[smoc-1(jj499 jj276)]*, or SMOC-1(M1+M2) *[smoc-1(jj499 jj511 jj276)]*, showing the long body size of SMOC-1(M1) worms, as compared to SMOC-1(WT) and SMOC-1(M1+M2) worms. Scale bar represents 50 μm. (**E, F**) Relative body sizes of various strains at the same developmental stage (WT set to 1.0). Groups marked with distinct symbols are significantly different from each other ($P < 0.001$, in all cases when there is a significant difference), while groups with the same symbol are not. The following are exceptions: the *P*-value between *jj499 jj510* and *jj276* is 0.003, while the *P*-value between *jj499 jj510* and *jj499 jj511* is 0.63. Tested using an ANOVA with a Tukey HSD. WT: $N = 22$. *tm7125*: $N = 33$. *jj276*: $N = 37$. *jj276/+* heterozygotes: $N = 29$. *jj499*: $N = 37$. *jj500*: $N = 32$. *jj499/+* heterozygotes: $N = 19$. *lon-2(e678)*: $N = 30$. *jj499 jj510*: $N = 32$. *jj499 jj511*: $N = 27$. Original data sets are in S1 Data. Original images of western blots can be found in S1 Raw Images. co-IP,

coimmunoprecipitation; EC, extracellular calcium-binding; IB, immunoblot; IP, immunoprecipitation; SMOC, secreted modular calcium-binding protein; WT, wild type.

glypican, does not function in BMP signaling [36]. In contrast to LON-2, SMOC-1 can positively promote BMP signaling when overexpressed ([12], and this study). Our co-IP results and in silico structural modeling showed that SMOC-1 can bind both LON-2 and the mature domain of DBL-1 and that SMOC-1 can mediate the formation of a LON-2-SMOC-1-DBL-1 tripartite complex. Furthermore, site-specific mutations in the endogenous *smoc-1 (smoc-1 (M1))* or the endogenous *lon-2 (lon-2(mut))* loci that disrupt SMOC-1-LON-2 interaction give the worms a long phenotype, while compound mutations in *smoc-1 (smoc-1(M1+M2))* that simultaneously disrupt SMOC-1-LON-2 interaction and weaken SMOC-1-DBL-1 interaction significantly shorten the length of *smoc-1(M1)* mutants to near WT length. Based on these data, we propose the following model for how SMOC-1 functions to regulate BMP signaling in *C. elegans* (Fig 12). In our model, SMOC-1 has both a negative, LON-2/glypican-dependent

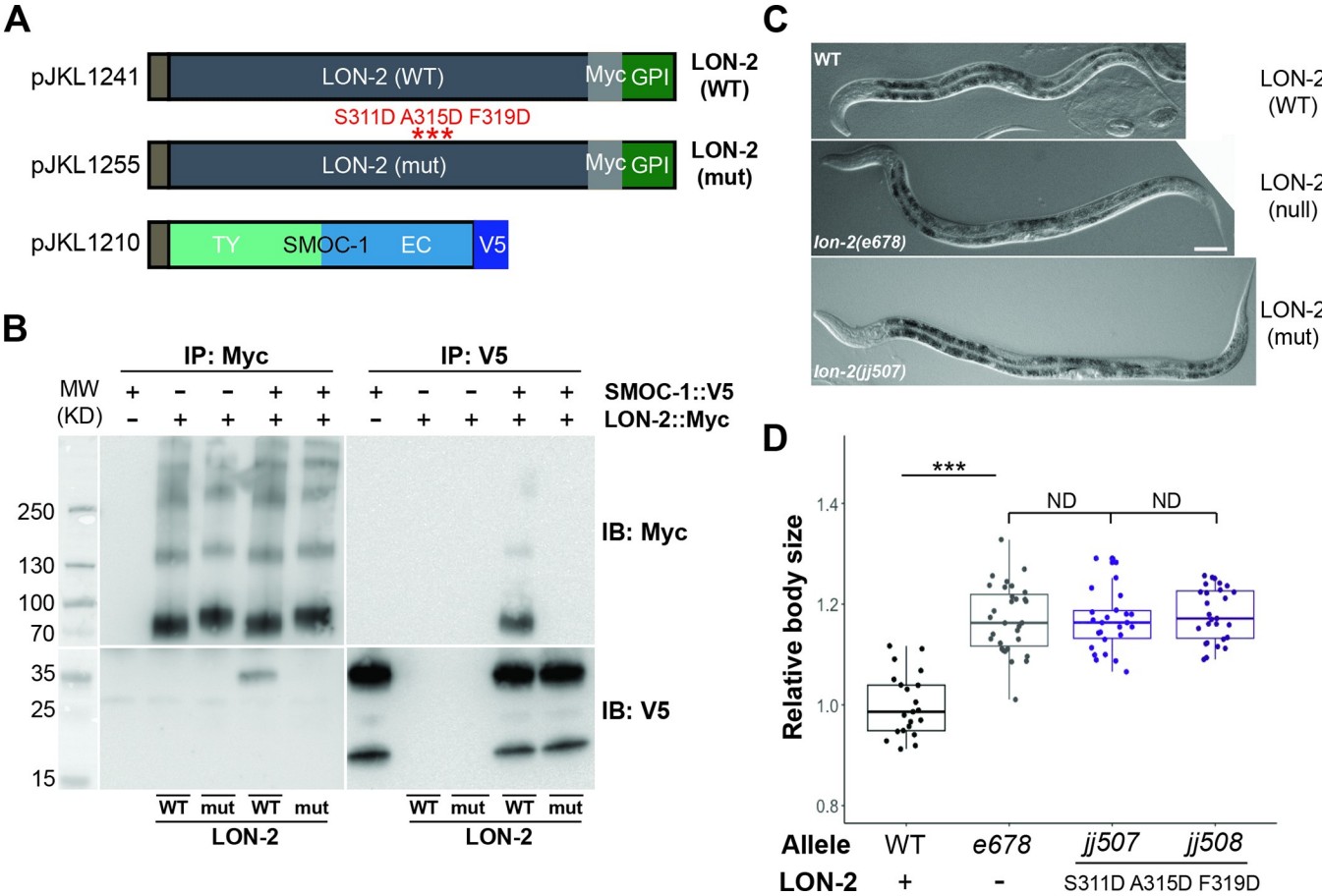

**Fig 10. The in vitro and in vivo consequences of mutations in LON-2 that affect SMOC-1 binding.** (**A**) Diagrams of SMOC-1, WT, and mutant LON-2 expression constructs used in the *Drosophila* S2 cell expression system. (**B**) Results of co-IP experiments testing the interaction between SMOC-1::V5 and Myc tagged WT and mutant LON-2 proteins. IP with anti-V5 beads or anti-Myc beads and IB with anti-V5 or anti-Myc antibodies, as indicated. Experiments were independently repeated in triplicate, with representative results shown in this figure. (**C**) DIC images of WT, *lon-2(jj678)* null, and *lon-2(jj507)* worms, showing their body sizes. Scale bar represents 50 µm. (**D**) Relative body sizes of worms at the same developmental stage (WT set to 1.0). ***$P < 0.001$; ND: no difference (ANOVA followed by Tukey HSD). WT: $N = 21$. *e678*: $N = 31$. *jj507*: $N = 26$. *jj508*: $N = 27$. Original data sets are in S1 Data. Original images of western blots can be found in S1 Raw Images. co-IP, coimmunoprecipitation; EC, extracellular calcium-binding; IB, immunoblot; IP, immunoprecipitation; SMOC, secreted modular calcium-binding protein; TY, thyroglobulin type-1; WT, wild type.

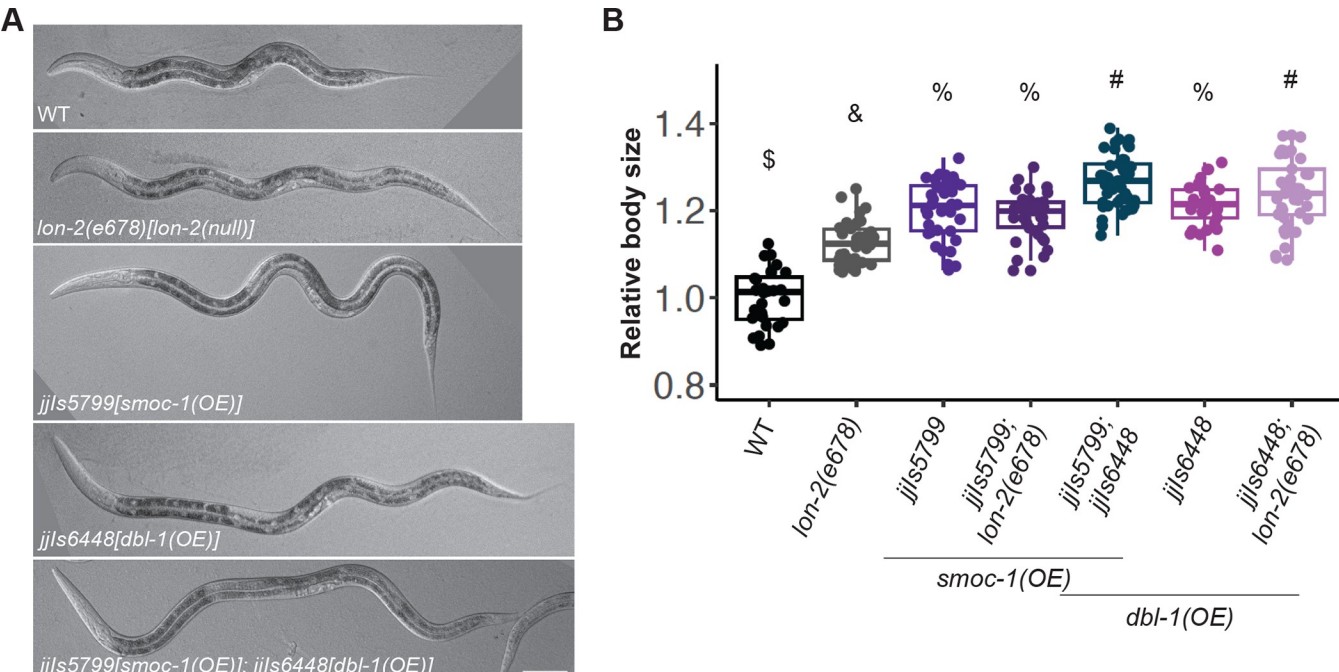

**Fig 11. Overexpression of SMOC-1 and DBL-1 has an additive effect on body size.** (**A**) DIC images of worms of various genotypes, including *lon-2(0)* null, *smoc-1(OE)*, *dbl-1(OE)*, and *smoc-1(OE); dbl-1(OE)* worms. Scale bar represents 50 μm. (**B**) Relative body sizes of various strains at the same developmental stage (WT set to 1.0). *lon-2(e678)* is a null allele of *lon-2*. *jjIs5799* is an integrated multicopy array line that overexpresses SMOC-1::2xFLAG. *jjIs6448* is an integrated multicopy array line that overexpresses DBL-1. For each of the genotypes except for WT, *e678* and *jjIs5799*, 2 independent isolates of the same genotype were measured, and data were combined for statistical analysis. Groups marked with distinct symbols are significantly different from each other ($P < 0.001$, in all cases when there is a significant difference), while groups with the same symbol are not. The following are exceptions: the *P*-value between *jjIs6448* and *jjIs6448; e678* is 0.55, while the *P*-value between *jjIs5799; e678* and *jjIs6448; e678* is 0.007. Tested using an ANOVA with a Tukey HSD. WT: $N = 28$. *e678*: $N = 37$. *jjIs5799*: $N = 39$. *jjIs5799; e678*: $N = 34$. *jjIs6448*: $N = 29$. *jjIs5799; jjIs6448*: $N = 45$. Original data sets are in S1 Data. SMOC, secreted modular calcium-binding protein; WT, wild type.

role and a positive, DBL-1/BMP-dependent but LON-2-independent role in regulating BMP signaling. On the one hand, SMOC-1 simultaneously binds to both the mature domain of DBL-1/BMP and LON-2/glypican, resulting in the sequestration of DBL-1/BMP to prevent DBL-1/BMP movement or interaction with the BMP receptors, thus inhibiting BMP signaling. On the other hand, secreted SMOC-1 can bind to the mature domain of DBL-1/BMP, possibly facilitating the movement of DBL-1/BMP through the extracellular space or the delivery/presentation of DBL-1/BMP to its receptors, thus promoting BMP signaling. The duality of SMOC-1 function in our model is consistent with previous findings that *smoc-1(0)* mutants are only slightly (approximately 5%) smaller than WT worms and that *smoc-1(0); lon-2(0)* double mutants have an intermediate body size between *smoc-1(0)* and *lon-2(0)* mutants [12]. It is also consistent with overexpression studies we presented in Fig 11, where overexpression of SMOC-1 or DBL-1 can promote BMP signaling in the absence of LON-2.

The concept of SMOCs acting as both antagonists and expanders of BMP signaling has been proposed by Thomas and colleagues [17]. *Xenopus* XSMOC-1 and the *Drosophila* SMOC homolog Pent have been shown to promote the spreading of the BMP ligand, thus expanding the range of BMP signaling, both in vitro and in vivo [10,11,17,37]. Both XSMOC-1 and human SMOC-1 can bind to heparin and heparan sulfate (HS), and in the case of *Drosophila* Pent, the BMP co-receptors Dally/glypican and Dally-like/glypican [10,17,27,28]. Studies in *Drosophila* further showed that the ability of Pent to extend the range of BMP signaling is dependent on Dally/glypican in wing imaginal discs [10]. Because neither XSMOC-1 nor Pent

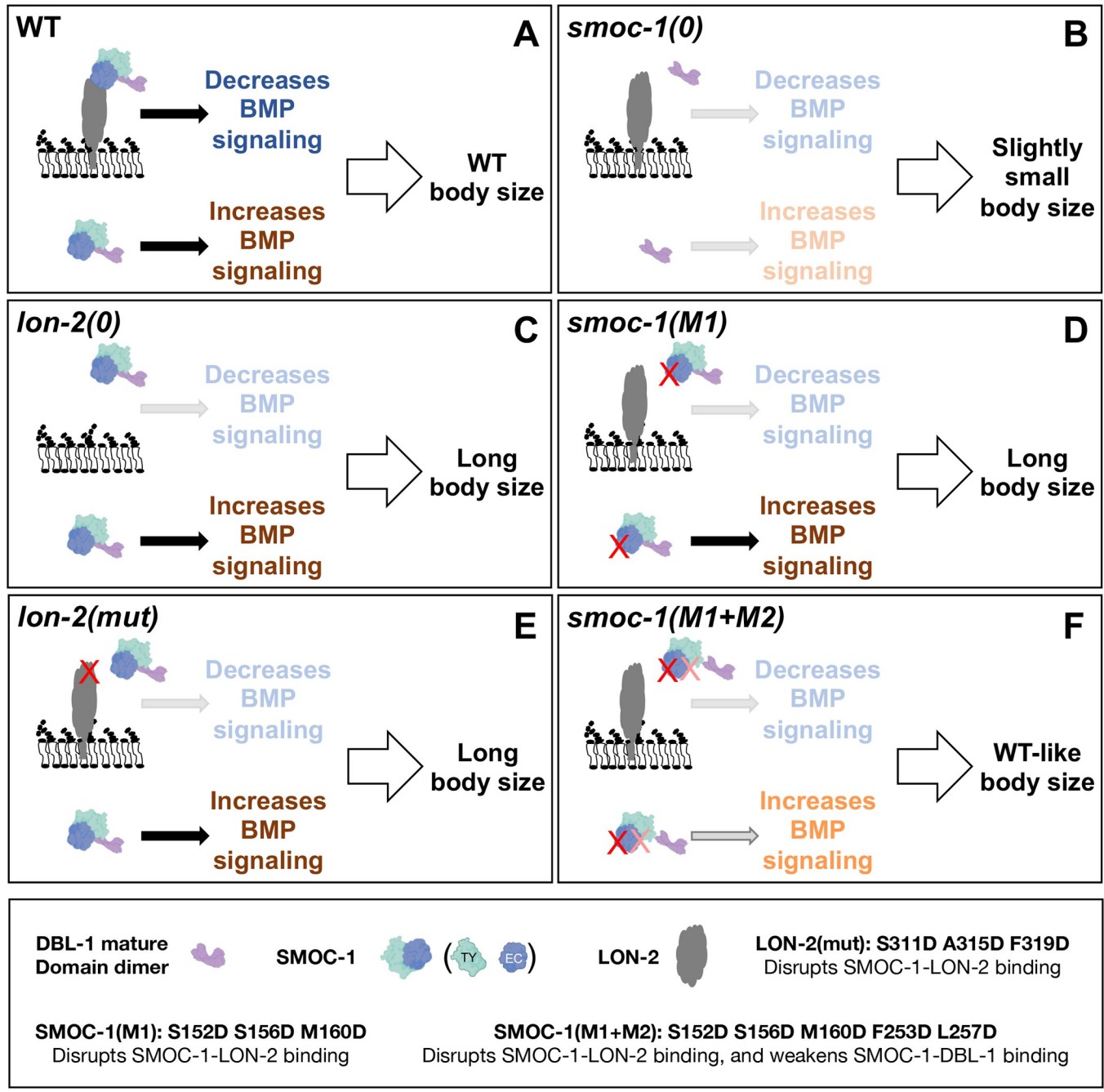

**Fig 12. A model for how SMOC-1 functions to regulate BMP signaling.** (**A**) In WT animals, SMOC-1 is proposed to both negatively and positively regulate BMP signaling. For its negative role, SMOC-1 binds to LON-2/glypican and DBL-1/BMP to sequester DBL-1/BMP. In the meantime, SMOC-1 can also bind to DBL-1/BMP to promote BMP signaling. The positive and negative roles of SMOC-1 in WT animals balance each other out, leading to a WT body size. (**B**) In *smoc-1(0)* animals, both the positive and negative roles of SMOC-1 are gone, leading to a slightly small body size. (**C–E**) Removing LON-2 completely [as in *lon-2(0)*] or disrupting SMOC-1-LON-2 interaction [as in *smoc-1(M1) or lon-2(mut)*] eliminates the negative role of SMOC-1, leading to a long body size. (**F**) Weakening SMOC-1-DBL-1 interaction in the SMOC-1(M1) background [as in *smoc-1(M1+M2)*] significantly attenuates the positive role of SMOC-1, leading to a non-long, WT-like body size. BMP, bone morphogenetic protein; SMOC, secreted modular calcium-binding protein; WT, wild type.

have been found to bind BMPs, but BMPs can bind HSPGs [10,17,19,28], a model was proposed where the binding of SMOCs to HSPGs competitively reduces the interaction between HSPGs and BMPs, thus promoting the spreading of BMPs [17]. Our data, combining in silico

structural modeling, in vitro biochemical assays, and in vivo site-directed mutagenesis, strongly suggest that the positive role for CeSMOC-1 in regulating BMP signaling is independent of LON-2/glypican, but dependent on DBL-1/BMP. This is consistent with our previous finding that *smoc-1(OE); dbl-1(0)* double mutants are as short as *dbl-1(0)* null mutants [12], suggesting that CeSMOC-1 acts through DBL-1/BMP to promote BMP signaling.

At present, there is no unified model on how SMOCs function to inhibit BMP signaling. Using a *Xenopus* animal cap assay, Thomas and colleagues [16,17] showed that both XSMOC-1 and Pent can inhibit BMP signaling downstream of the BMP receptor, at least for XSMOC-1, by activating the MAPK pathway. In contrast, *Drosophila* Pent has been shown to function upstream of receptor activation to inhibit BMP signaling in zebrafish [10]. A more recent study showed that mouse SMOC2 can inhibit BMP signaling by competitively binding to BMPR1B [13]. Our results are consistent with CeSMOC-1 functioning in a LON-2/glypican-dependent manner to negatively regulate BMP signaling.

## Glycosaminoglycan (GAG)-dependent and -independent interactions between CeSMOC-1 and LON-2/glypican

Using both IP-MS of worm extracts and coimmunoprecipitation of proteins expressed in *Drosophila* S2 cells, we have found that CeSMOC-1, specifically the EC domain of CeSMOC-1, can bind to LON-2/glypican. Our findings are consistent with previous findings showing that *Drosophila* Pent can co-IP with Dally/glypican and Dally-like/glypican, while XSMOC-1, hSMOC2, and Pent can all bind to heparin and HS in vitro [10,17,27,28]. The EC domains of SMOCs all have a stretch of positively charged residues that was hypothesized to mediate its interaction with HSPGs [27]. Fascinatingly, our protein structural modeling suggests that the SMOC EC domain may directly interact with the protein core of HSPGs. As shown in Fig 8, the key residues identified to mediate SMOC-1-LON-2 interaction are conserved among nematode SMOC proteins, as well as vertebrate SMOCs. Further, our results in *C. elegans* provide additional clarity to some previously perplexing findings by Taneja-Bageshwar and Gumienny [38,39]. These authors showed that overexpressing the LON-2/glypican protein core is sufficient to rescue the long (Lon) body size phenotype of *lon-2 (0)* null mutants, suggesting that the LON-2/glypican protein core is sufficient to inhibit BMP signaling. Their previous findings further showed that the N-terminal region of LON-2, LON-2(1–368), is sufficient to inhibit BMP signaling. Notably, this region contains all the key residues in LON-2 that mediate LON-2-SMOC-1 interaction, as identified by our structural modeling (between amino acids 100 and 325, S4 Fig). Mutating the 3 amino acids in LON-2 predicted to mediate LON-2-SMOC-1 interaction, as in LON-2(mut) [S311D A315D F319D], resulted in mutant worms exhibiting a long body size phenotype indistinguishable from that of *lon-2(0)* null mutants (Fig 10). These results strongly argue that the major, if not the sole, function of LON-2 in regulating body size is mediated by its interaction with SMOC-1 via the LON-2 protein core.

In the *C. elegans* study mentioned above, Taneja-Bageshwar and Gumienny [38,39] also showed that the HS attachment sites are important for a truncated form of LON-2 (aa423-508) to function in inhibiting BMP signaling when overexpressed. Whether similar results hold true in the endogenous *lon-2* locus remains to be determined. Kirkpatrick and colleagues found that in *Drosophila*, the protein core of Dally/glypican is partially functional, while its overexpression actually limits Dpp/BMP signaling in the wing imaginal discs [40]. Whether Pent is involved in mediating this function of Dally/glypican is unclear.

## CeSMOC-1 as a mediator of glypican and BMP interaction

LON-2/glypican is a well-established negative regulator of BMP signaling in *C. elegans* [35]. Previous studies showed that when expressed in mammalian HEK293T cells, membrane-

tethered LON-2 can bind to mammalian BMP2 after chemical crosslinking [35]. Since *Drosophila* and vertebrate BMPs have been found to bind to HSPGs [19,41,42], a model was proposed where LON-2/glypican negatively regulates BMP signaling by sequestering the DBL-1/BMP ligand in *C. elegans* [35]. We did not detect any interaction between LON-2 and any form of DBL-1, either by co-IP assays using the *Drosophila* S2 cell system, or via in silico structural modeling (Figs 5, S2 and S3). Although we cannot rule out the possibility that posttranslational modifications happening in *Drosophila* S2 cells are different from those found in *C. elegans*, we believe that the absence of LON-2 and DBL-1 interaction in our S2 cell expression system is unlikely due to the lack of glycosaminoglycan (GAG) modifications of LON-2. As shown in Figs 2, 5 and S2, we detected multiple higher molecular weight bands on western blots that likely correspond to GAG-decorated LON-2.

Our results strongly argue that LON-2/glypican binds DBL-1/BMP indirectly via CeSMOC-1 as a bridging molecule. We detected strong interaction between CeSMOC-1 and the mature domain of DBL-1/BMP (Fig 5) and showed that CeSMOC-1 can mediate the formation of a tripartite complex between LON-2/glypican and DBL-1/BMP (Fig 8). In silico structural modeling using the ColabFold [34] implementation of AlphaFold2 [30] supported the interactions between LON-2/glypican, CeSMOC-1, and mature DBL-1/BMP (Figs 6–8). Importantly, key amino acids identified by structural modeling in each of the 3 proteins to mediate pair-wise protein–protein interactions appear to be highly conserved (Figs 6, S3 and S4). Finally, mutations in CeSMOC-1 or LON-2/glypican that disrupt SMOC-1-LON-2 interaction (as in SMOC-1(M1) or LON-2(mut)) resulted in a long body size phenotype, a manifestation of increased BMP signaling, while mutations that significantly weaken CeSMOC-1-DBL-1/BMP interaction in the SMOC-1(M1) background, as in SMOC-1(M1+M2), abolished the long phenotype of SMOC-1(M1) mutant worms (Figs 9 and 10). Our data suggest that LON-2/glypican acts as a negative regulator of BMP signaling in *C. elegans* by sequestering the DBL-1/BMP ligand via CeSMOC-1 and that CeSMOC-1 facilitates the negative regulation of BMP signaling by forming a tripartite complex with LON-2/glypican and DBL-1/BMP. Since SMOC homologs from *C. elegans*, *Drosophila*, and *Xenopus* all functionally interact with HSPGs, we propose that the SMOC-HSPG axis is an evolutionarily conserved module important in regulating BMP signaling.

## SMOC-1-BMP interaction in promoting BMP signaling

Our previous work [12] and this work showed that when overexpressed, CeSMOC-1 can act through DBL-1/BMP to positively promote BMP signaling. We argue that this positive role of CeSMOC-1 is LON-2/glypican-independent and primarily mediated by direct interactions between CeSMOC-1 and DBL-1/BMP. This is strongly supported by results from our in vitro biochemical assays, in vivo mutagenesis studies, and predictions from in silico structural modeling. Direct interaction between SMOCs and BMPs has not been previously reported in any organisms. SMOC proteins from different organisms also contain different numbers of the TY or EC domains in different arrangement (Fig 7A). Nevertheless, our structural modeling predicted with high confidence direct interactions between SMOC and BMP homologs in *H. sapiens*, *X. laevis*, and *D. melanogaster* (Fig 7B). The predicted interactions also share similarities with the predictions for CeSMOC-1 and DBL-1/BMP. We therefore propose that the interaction between SMOCs and BMPs is evolutionarily conserved.

At present, it is not clear how CeSMOC-1 can promote BMP signaling by binding to DBL-1/BMP. It is possible that CeSMOC-1 could facilitate either the movement of DBL-1/BMP through the extracellular space or the binding of DBL-1/BMP to its receptors, thus promoting BMP signaling. Whether CeSMOC-1 accomplishes this role alone or with the help of another

protein(s) is currently unknown. Future research will test these hypotheses and determine the temporal and spatial specificity of CeSMOC-1 function in regulating BMP signaling.

## The importance of both the TY and the EC domains for SMOC-1 function

Previous studies on SMOC homologs have shown that the EC domain of SMOC proteins can bind to HSPGs. In this study, we showed that the EC-mediated interaction between CeSMOC-1 and LON-2/HSPG is essential for the inhibitory functions of CeSMOC-1 and LON-2/glypican in regulating BMP signaling. We further showed that both the TY and the EC domains are required for full function of CeSMOC-1 at the endogenous locus. The requirement for both the TY and the EC domains for full CeSMOC-1 function is consistent with both of our co-IP and structural modeling results, which demonstrated that residues in both the TY and the EC domains are involved in the interaction between CeSMOC-1 and DBL-1/BMP (Figs 5, 6 and 9). However, paradoxically, overexpression of SMOC-1(EC) is sufficient to promote BMP signaling in vivo (Fig 3), even though SMOC-1(EC) did not bind DBL-1/BMP in our in vitro co-IP assays (Fig 2). It is possible that the stringent IP condition that we used precludes possible low-affinity interactions between SMOC-1(EC) and DBL-1/BMP that can occur in *C. elegans*, especially when SMOC-1(EC) is drastically overexpressed. Consistent with this notion, we have previously shown that 2 single amino acid substitution mutations in the respective TY domain and EC domain of SMOC-1, *jj85(E105K)* and *jj65(C210Y)*, caused partial loss-of-function phenotypes at the native environment [12]. Yet, like overexpressing SMOC-1(EC), overexpressing either SMOC-1(E105K) or SMOC-1(C210Y) was also sufficient to promote BMP signaling by increasing the body size (S8 Fig). Since many functional assays in *C. elegans* involve the utilization of repetitive transgenic arrays that can cause overexpression of the transgene, our results also highlight the importance of carrying out functional studies of proteins at the endogenous expression levels.

In this study, we showed that key residues in the TY domain of *C. elegans* SMOC-1 mediate CeSMOC-1-DBL-1/BMP interaction, as mutating these key residues (as in SMOC-1(M3) [Y72A Y90A Y95A W97A]) significantly compromised the ability of CeSMOC-1 to bind DBL-1/BMP in vitro (Fig 9). Our structural modeling suggested that the TY domains in SMOC proteins from *C. elegans* to humans are also involved in mediating SMOC-BMP interaction (Fig 7). Interestingly, Thomas and colleagues found that one of the TY domains (TY1) in XSMOC-1 is required for XSMOC-1's function in inhibiting BMP signaling [17]. Based on our findings, we speculate that perhaps this role of XSMOC-1 TY1 in BMP signaling is mediated via its direct interaction with BMP.

The TY domain is evolutionarily conserved across metazoans and seen in a variety of functionally diverse proteins, including thyroglobulin, SMOCs, nidogens, and IGFBPs [43,44]. The TY (type 1a) domain is characterized by 6 conserved cysteine residues, including "QC" and "CWCV" residue motifs, that form intramolecular disulfide bridges [45]. Outside of these conserved regions, the TY domain has highly variable loops that may contribute to the wide variety of associated functions. TY domain-containing proteins may function to inhibit cysteine cathepsins, while others inhibit aspartate peptidases, papain-like proteases, or metalloproteases [43,44]. Testicans, which contain a TY domain as well as an EC domain, can work as a competitive inhibitor of cathepsin L as found in human cell culture. On the other hand, human testicans have also been found to bind to cathepsin L to act as a chaperone, preventing degradation and encouraging cathepsin activity within the ECM [46,47]. Future research will determine whether the TY domains in these other proteins can function analogously to the TY domain in CeSMOC-1 by binding to members of the TGFβ super family of growth factors and whether SMOC TY domains have additional functions beyond TGFβ signaling.

## Conclusion

There are 2 SMOC homologs in mammals, SMOC1 and SMOC2. Both mice and human individuals with mutations that severely reduce SMOC1 expression exhibit abnormal tooth, eye, and limb development [14,15,48–50]. Mutations in hSMOC2 have been found to be associated with defects in tooth development [51,52] and vitiligo [14,53,54]. Abnormal expression of SMOCs has also been associated with multiple cancers as well as kidney and pulmonary fibrosis [55–61]. Our work described in this study identified evolutionarily conserved mechanisms on how SMOC proteins regulate BMP signaling. We have previously shown that both hSMOC1 and hSMOC2 can partially rescue the BMP signaling defect of *smoc-1(0)* null mutants in *C. elegans* [12]. Future research on the mechanistic details of SMOC function to regulate BMP signaling in an in vivo system such as *C. elegans* may have significant implications for human health.

## Materials and methods

### Plasmid constructs and transgenic lines

All plasmids and oligonucleotides used in this study are listed in S4 and S5 Tables, respectively. Plasmids used for expression in *Drosophila* S2 cells were generated from pCB313 (gift from Dr. Claire Bénard, [62]).

Transgenic strains were generated using pCFJ90[*myo-2p::mCherry::unc-54 3'UTR*] (gift from Erik Jorgensen) or LiuFD290[*ttx-3p::mCherry*] (gift from Oliver Hobert) as co-injection markers. Two transgenic lines with the best transmission efficiency across multiple generations were analyzed with each plasmid. Integrated transgenic lines were generated using gamma irradiation, and then outcrossed with WT N2 worms at least 2 times. S6 Table lists all the strains generated in this study.

For IP/MS experiments, integrated transgenic lines overexpressing untagged SMOC-1 or SMOC-1::2xFLAG were used (see S6 Table). Transgenic strains overexpressing either SMOC-1(TY)::2xFLAG or SMOC-1(EC)::2xFLAG were generated using an embryonic lethal temperature-sensitive *pha-1(e2123)* background, by including a *pha-1* expressing plasmid (pC1) and a florescent co-injection marker within the array. Only transgenic worms are viable at the restrictive temperature (25˚C) [63]. The florescent co-injection marker allowed for visual assessment of multi-generational transgene transmission efficiency. The usage of these 2 markers allowed us to grow large populations of exclusively transgenic worms carrying the transgenes at high transmission efficiency.

### Generating endogenous *smoc-1::2xflag*, *smoc-1(TY)::2xflag*, *smoc-1 (EC)::2xflag*, and *HA::dbl-1* strains using CRISPR/Cas9-mediated homologous recombination

All sgRNA guide plasmids were generated using the strategy described in [64,65]. To generate each specific modifications, the specific sgRNA plasmids (see S4 Table) were injected with the Cas9-encoding plasmid, pDD162 (Dickinson and colleagues), and the single-strand oligodeoxynucleotide homologous repair template, into WT N2 worms. pRF4(*rol-6(d)*) was used as a co-injection marker. Injected P0 animals were singled. F1 progeny were picked from plates that gave the most roller progeny, allowed to lay eggs and screened for successful CRISPR events via PCR (see S5 Table for oligo information). The resulting stable strains were confirmed by genotyping and Sanger sequencing. The *smoc-1(TY)::2xflag* and the *smoc-1(EC)::2x-flag* strains were generated in the *smoc-1(jj276[smoc-1::2xflag])* background using either

single-strand oligodeoxynucleotide or plasmid as homologous repair template (see S4 and S5 Tables).

## Generating endogenous *smoc-1(M1)::2xflag*, *smoc-1(M1+M2)::2xflag*, and *lon-2(M)* strains using CRISPR/Cas9-mediated homologous recombination

CRISPR–Cas9-mediated mutations in the *smoc-1* or *lon-2* locus were generated using ribonucleoprotein (RNP) complexes by following the protocol described in [66]. RNP mixes containing 0.25 mg/ml Cas9 (IDT 1081059), 0.1 mg/ml tracrRNA (IDT 1072534), 56 ng/ml gene-specific crRNA (IDT), 25 ng/ml single-stranded oligodeoxynucleotide repair template (IDT ultramers), and 40 ng/ml of pRF4(*rol-6(d)*) plasmid were injected into either the *smoc-1(jj276 [smoc-1::2xflag])* (for *smoc-1* edits) or PD1074 WT (for *lon-2* edits) worms. Injected P0 animals were singled. F1 progeny were picked from plates that gave the most roller progeny, allowed to lay eggs and screened for successful CRISPR events via PCR. See S5 Table for oligo information. The resulting stable strains were confirmed by genotyping and Sanger sequencing of the entire *smoc-1* or *lon-2* coding region.

## Generating strains containing integrated extra-chromosomal arrays overexpressing either *smoc-1::2xflag* or *HA::dbl-1*

Genomic DNA including the coding region of *smoc-1::2xflag*, 2 kb upstream and 2 kb downstream sequences was amplified from *jj276* worms using JKL1549 and JKL1550 and cloned into a pBSII SK+ vector to generate the plasmid pMSD35. Transgenic strains were generated using *ttx-3p::mCherry* (LiuFD290) as a co-injection marker. Integrated transgenic lines (*jjIs5798*, *jjIs5799*, *jjIs5800*) were generated using gamma-irradiation, followed by 2 rounds of outcrossing with N2 worms. Similar approaches were used to generate pTYC3, which has sequences corresponding to the genomic DNA that includes the coding region of *HA::dbl-1*, 3 kb upstream and 0.7 kb downstream sequences, cloned into a pBSII SK+ vector. Transgenic strains were generated using pCFJ90[*myo-2p::mCherry::unc-54 3'UTR*] as a co-injection marker. A spontaneously integrated strain (*jjIs6448*) was generated, and then outcrossed for 3 rounds with WT worms.

## Body size measurements

Body size measurements were conducted as previously described [12,67]. Gravid adults were bleach synchronized, with resulting embryos incubated in M9 buffer rotating at 15˚C until hatched (24 to 48 h). Synchronized L1s were plated and grown at 20˚C until the L4.3 vulval stage was seen in a majority of worms. To measure the body length of heterozygous *jj276/+* or *jj499 jj276/+* hermaphrodites, *jjIs3900/+* males that express mCherry in the pharynx (S6 Table) were mated with *jj276* or *jj499 jj276* hermaphrodites, red cross progeny at the L4.3 vulval stage were collected for measurement. For imaging, worms were washed from plates, treated with 0.3% sodium azide until straightened, and then mounted onto 2% agarose pads. Images were taken using a Hamamatsu Orca-ER camera using the iVision software (Biovision Technology). Using Fiji, worm body lengths were measured from images using the segmented line tool. An ANOVA and Tukey's honest significant difference (HSD) was used to test for differences in body size between genotypes using R (https://www.R-project.org/).

## Suppression of *sma-9(0)* M-lineage defect (Susm) assay

For the suppression of *sma-9(0)* M-lineage defect (Susm) assay, worms were grown at 20˚C, and the number of adult animals with 4 CCs and 5 to 6 CCs were tallied across multiple plates

[68]. The reported Susm penetrance refers to the percent of animals with 1 or 2 M-derived CCs as scored using the CC::GFP reporter. For each genotype, 2 independent isolates were generated (as shown in the strain list in S6 Table), 3 to 7 plates of worms from each isolate were scored for the Susm phenotype, and the Susm data from the 2 isolates were combined, and the means of the Susm penetrance were presented. The lack of M-derived CCs phenotype is not fully penetrant in *sma-9(cc604)* mutants [28]. For the Susm rescue experiments, we used R to generate a general linear model with binomial error and a logit link function designating transgenic state as the explanatory function. The Wald statistic test was used to determine if transgenic state (transgenic versus non-transgenic worms within the same line) is associated with CC number.

## Immunoprecipitation of FLAG-tagged proteins from *C. elegans*

*C. elegans* strains were repeatedly bleach synchronized and grown on 90 mm NGM plates seeded with NA22 bacteria, until desired population size was reached. Approximately 25 NA22 plates containing about 10,000 synchronized gravid adults were bleached to get a target population of about 2 million or more synchronized L1s in M9. Approximately 4,000,000 synchronized L1s were plated on 15 cm egg plates (NGM strep with OP50-1 and chicken egg; [66] and grown until population reached the L4 stage (48 h at 25˚C)). Worms were washed from plates and collected with H150 (50 mM HEPES (pH 7.6), 150 mM KCl). Successive pelleting and washing were done to remove any excess food or debris. Finally, worms were pelleted and an equal volume of H150g10 (H150 with 10% glycerol) with protease inhibitor (Pierce, A32965) was added to make a worm slurry. "Worm popcorn" was made by adding the slurry dropwise directly into liquid nitrogen. Resulting popcorn was stored in 50 mL conical tubes at −80˚C.

To physically break worms, a mortar and pestle was used to grind the popcorn until no-to-few intact worms were visible. For the first IP/MS experiment, 20 g of popcorn was used per strain, while 10 g of popcorn was used in the second IP/MS experiment. Samples were kept in liquid nitrogen throughout the process to avoid unwanted thawing. Worm homogenates were then thawed on ice and diluted 1:5 (w:v) with H150g10 with 1% Triton. Samples were centrifuged at 12,000g at 4˚C to separate soluble and insoluble fractions. Soluble fraction was filtered using a 0.45 μm filter (Fisher brand Disposable PES Bottle Top filters, FB12566511) before being added to pre-equilibrated anti-Flag M2 magnetic beads (Millipore Sigma, M8823) for incubation overnight at 4˚C with rotation. After incubation, unbound fraction was removed and beads were washed 3 times with H150g10 to remove unbound proteins. A final wash in TBS (20 mM Tris-HCl, 150 mM NaCl (pH 7.6)) was done before eluting with FLAG peptide (Sigma, F3290). Elution was done by adding 5× volumes of packed bead volume of FLAG peptide in TBS at 150 ng/μl, followed by incubation at 4˚C for 30 min. This was repeated for each sample and the 2 eluates were pooled together.

## Mass spectrometry

Eluates were submitted to Biotechnology Resource Center (BRC) at Cornell University for analysis by MS. Briefly, samples were prepared by in-solution trypsin digestion before conducting nanoLC-ESI-MS/MS analysis using an Orbitrap Fusion Tribrid (Thermo Fisher Scientific, San Jose, California, United States of America) mass spectrometer equipped with a nanospray Flex Ion Source and coupled with a Dionex UltiMate 3000 RSLCnano system (Thermo, Sunnyvale, California, USA) [69,70]. Processing workflow used SequestHT and MS Amanda with Percolator validation. Database search was conducted against a *C. elegans* database downloaded from NCBI in June 2021. Only high confidence peptides defined by Sequest

HT with a 1% FDR by Percolator were considered for confident peptide identification. Abundance ratios relative to untagged (e.g., *smoc-1*::*2xflag/smoc-1*) were assessed to identify candidate interaction partners, with values over 2.0 being considered enriched. Only hits with 2 or more mapped peptides were considered here.

## Coimmunoprecipitation of proteins expressed in *Drosophila* S2 cells

*Drosophila* S2 cells were grown in M3+BPYE+10% Heat-inactivated FBS and transfected using a calcium phosphate method (Invitrogen protocol, Version F 050202 28–0172). For non-secreted proteins, such as LON-2, which is GPI-anchored, or for harvesting full-length DBL-1 prior to its secretion, cells were collected 2 days post-transfection and lysed in lysis buffer (50 mM Tris (pH 7.6), 150 mM NaCl, 1 mM EDTA, 1% Triton-X). For secreted proteins, such as SMOC-1 and the processed forms of DBL-1, cell media was collected 5 to 7 days post-transfection. Protease inhibitor (Pierce, A32965) was added to all samples to avoid protein degradation. Westerns were conducted to confirm and roughly evaluate protein levels. Samples were stored at −80°C until use.

Anti-HA (EZView red Anti-HA affinity gel, Sigma 45-E6779), anti-V5 (Anti-V5 agarose affinity gel, Sigma 45-A7345), EZView red anti-c-Myc affinity gel (Sigma E6654), and EZView red Anti-FLAG beads (Sigma F2426) were used for IP of target proteins. In each trial, the 2 lysates (or media) each containing a protein of interest (POI) were mixed together before being added to appropriate beads. Single protein controls were also applied to beads to assess for any nonspecific binding. Samples were rotated on beads overnight at 4°C. The following day, unbound protein was removed. Five successive washes using wash buffer (50 mM Tris-HCl (pH 7.5), 150 mM NaCl, 1% Triton X-100) were done to remove any additional unbound proteins, with centrifugation to pellet beads between each wash. Bound proteins were eluted by the addition of elution buffer (100 mM Tris-HCl (pH 8.0), 1% SDS), followed by removal of beads using spin filters. Resulting eluate samples were prepared by the addition of 5× SDS buffer (0.2 M Tris-HCl (pH 6.8), 20% glycerol, 10% SDS, 0.25% bromophenol blue, 10% β-mercaptoethanol) and a 10-min incubation at 95°C.

## Preparation of worm lysates for SDS-PAGE and western blot analysis

Western blot of *C. elegans* was conducted to detect proteins expressed in vivo. Indicated number of worms were picked into 20 μl of double distilled water and lysed by addition of 5 μl of 5xSDS buffer (0.2 M Tris-HCl (pH 6.8), 20% glycerol, 10% SDS, 0.25% bromophenol blue, 10% β-mercaptoethanol) followed by snap freezing in liquid nitrogen. Samples were heated to 95°C for 10 min, stored at −20°C, and used for subsequent SDS-PAGE and western blot analysis.

## Sodium dodecyl sulfate-polyacrylamide gel electrophoresis (SDS-PAGE) and western blotting

Proteins from worm lysates or Co-IP experiments were separated on 10% or 4% to 20% gradient Mini-PROTEAN TGX Precast Gels (Bio-Rad Laboratories) at 300 V. Proteins were transferred to Immobilon-P PVDF membrane (MilliporeSigma) using Power Blotter Station (Model: PB0010, Invitrogen by Thermo Fisher Scientific) for 7 min for 10% gels or 8 min for 4% to 20% gels at 1.3 A and 25 V. Membranes were blocked with EveryBlot Blocking Buffer (Bio-Rad Laboratories) for 5 min at room temperature. The resulting membranes were incubated with indicated primary antibody in Everyblot buffer at 4°C overnight with gentle shaking. The following day, membranes were washed in 1xPBST (137 mM NaCl, 2.7 mM KCl, 10 mM $Na_2HPO4$, 1.8 mM $KH_2PO4$, 0.1% Tween-20) for 10 min, repeated 3 times total.

Incubation with indicated secondary antibodies was done in Everyblot buffer or PBST + 5% dry milk (Carnation) for 2 h at room temperature. After 3 additional PBST washes, membranes were developed using Clarity ECL Reagent (Bio-Rad 1705061) and imaged using a Bio-Rad ChemiDoc MP imaging system. When needed, membranes were stripped using Restore western blot stripping buffer (Pierce), blocked again, and reblotted.

Primary antibodies used include mouse monoclonal ANTI-FLAG M2 antibody (diluted 1:5,000, F3165, Sigma), mouse anti-HA IgG monoclonal antibody (Krackler, 12CAS, diluted 1:1,000), rabbit anti-HA IgG monoclonal antibody (Cell Signaling, C29F4, diluted 1:1,000), mouse anti-V5 IgG monoclonal antibody (Invitrogen (E10/V4RR), diluted 1:1,000), rabbit anti-V5 IgG monoclonal antibody (Cell Signaling, D3H8Q, diluted 1:1,000), mouse anti-actin IgM JLA20 monoclonal antibody (Developmental Studies Hybridoma Bank; diluted 1:2,000), and anti-Myc monoclonal antibody 9E10 (diluted 1:40). Secondary antibodies used include horseradish peroxidase-conjugated donkey anti-mouse IgG, peroxidase-conjugated goat anti-rabbit IgG, horseradish peroxidase-conjugated goat anti-mouse IgM (all from Jackson ImmunoResearch; diluted 1:10,000), and IRDye 800CW goat anti-mouse IgM (Li-COR, diluted 1:5,000).

## In silico protein–protein interaction structure predictions

The ColabFold [34] implementation of AlphaFold ([30] was used to predict structures of complexes involving different combinations of LON-2, SMOC-1, and the mature form of DBL-1. Default parameters were used except 3 models were predicted for each query. Pairwise combinations resulted in predictions for an interaction between LON-2 and SMOC-1, with an interface predicted template modeling ("iptm") score of 0.59 and an interaction between SMOC-1 and mature DBL-1 with an iptm score of 0.76. In contrast, no interaction was predicted between mature DBL-1 and LON-2. Given these predictions, we then attempted a prediction with LON-2, SMOC-1, and 2 copies of mature DBL-1 (mature DBL-1 is known to homodimerize). The result was a structural prediction in which SMOC-1 interacts with both mature DBL-1 and LON-2 (iptm score of 0.5) through interfaces that are the same as predicted in the 2 corresponding pairwise combinations. In each of these successful predictions, the 3 models predicted for each case had identical predicted interfaces. For predictions of complexes formed between CeSMOC-1 and mature DBL-1 homologs in other species, the following iptm scores were obtained for the top hits: *C. elegans* SMOC-1:DBL-1:DBL-1, 0.76; *H. sapiens* SMOC1: BMP2:BMP2, 0.81; *X. laevis* XSMOC1:BMP2:BMP2, 0.78; *D. melanogaster* Pentagone:Dpp: DPP, 0.80. The iptm score is a confidence score generated by AlphaFold ([33]).

## Supporting information

**S1 Table. LON-2 peptides recovered from the IP-MS experiments.**
(PDF)

**S2 Table. Results of IP-MS experiment 1 showing proteins that are more than 10-fold enriched in the tagged (SMOC-1::2xFLAG(OE)) vs. untagged (SMOC-1(OE)) samples.**
(XLSX)

**S3 Table. Results of IP-MS experiment 2 showing proteins that are both enriched in the tagged (SMOC-1::2xFLAG(OE)) vs. untagged (SMOC-1(OE)) samples, as well as in the (SMOC-1(EC)::2xFLAG(OE)) vs. untagged (SMOC-1(OE)) samples.**
(XLSX)

**S4 Table. Plasmids generated in this study.**
(PDF)

**S5 Table. Oligonucleotides used in this study.**
(PDF)

**S6 Table. *C. elegans* strains used in this study.**
(PDF)

**S1 Fig. GFP-tagged SMOC-1 is non-functional.** (**A**) Diagrams depicting various endogenously tagged SMOC-1 proteins, with the corresponding CRISPR alleles shown on the left of the diagrams. (**B**) Table showing the penetrance of the Susm phenotype of strains carrying specified *smoc-1* allele in a *sma-9(cc604)* background. The Susm penetrance refers to the percent of animals with 1 or 2 M-derived CCs as scored using the *arIs37(secreted CC::GFP)* reporter. For each genotype, 2 independent isolates were generated (as shown in the strain list), 4 to 7 plates of worms from each isolate were scored for the Susm phenotype at 20°C, and the Susm data from the 2 isolates were combined and presented in the table. [a] The lack of M-derived CCs phenotype is not fully penetrant in *sma-9(cc604)* mutants [25]. [b] Data from [12]. Statistical analysis was conducted by comparing various double mutants with the *sma-9(cc604)* single mutant. **** $P < 0.0001$ (unpaired two-tailed Student's *t* test). (**C**) Relative body sizes of various strains at the same developmental stage (WT set to 1.0). Body sizes of 35 to 70 worms of each genotype were measured. A *dbl-1* null allele (*ok3749*) and *smoc-1* null allele (*tm7125*) were included as controls. Groups marked with distinct symbols are significantly different from each other ($P < 0.001$, in all cases when there is a significant difference, except that the *P*-value between *tm7125* and *jj271* is 0.0099), while groups with the same symbol are not. The exception is tested using an ANOVA with a Tukey HSD. Original data sets are in S1 Data.
(PDF)

**S2 Fig. LON-2 does not interact with full-length DBL-1.** (**A**) Diagrams of LON-2 and DBL-1 expression constructs used in the *Drosophila* S2 cell expression system. (**B**) Results of co-IP experiments testing the interaction between LON-2::Myc and V5::DBL-1 prodomain::FLAG:: DBL-1 mature domain. Immunoprecipitation (IP) with anti-Myc beads or anti-V5 beads and immunoblot (IB) with anti-Myc or anti-FLAG antibodies, as indicated. Full-length DBL-1 detected by anti-FLAG antibody runs at around 55KD. Experiments were independently repeated in triplicate, with representative results shown in this figure. Original images of western blots can be found in S1 Raw Images.
(PDF)

**S3 Fig. BMP ligands are not predicted to interact with glypicans.** Predicted alignment error (PAE) plots are shown for predictions involving BMP ligands produced by the ColabFold implementation of AlphaFold. PAE values are shown as a heat map, with blue representing low values and red representing high values. Blue represents residue pairs for which there is high confidence in their locations relative to each other in the 3D prediction. Red represents residue pairs with no confidence in their locations relative to each other in the 3D prediction. As indicated in the Key (top left), the regions of the plots within the dashed rectangles highlight the portions of the plot corresponding to predicted interactions between the BMP ligands and the protein of interest. The interaction between *H. sapiens* SMOC1 and the homodimeric BMP2 ligand is predicted with high confidence, as indicated by the overall low PAE values for the interacting residue pairs (top right). In contrast, the plots for *H. sapiens* glypican 1 with different combinations of BMP2/4 dimeric ligands, and for *C. elegans* LON-2 with the homodimeric DBL-1 ligand, show high predicted errors indicating a lack of predicted interaction. Similar negative results were observed for predictions involving full-length or prodomain

sequences of the BMP ligands.
(PDF)

**S4 Fig. Multiple sequence alignment of LON-2 homologs.** Clustal Omega (CLUSTAL O (1.2.4)) [71] of *C. elegans* (Ce) LON-2 with its homologs from other nematode species, including *C. Japonica* (Cjp), *C. brenneri* (Cbn), *C. briggsae* (Cbr), and *C. remanei* (Cre), as well as with *Drosophila* Dally and Glypican from *M. musculus* (mouse) and *H. sapiens* (human). Red $ marks the residues at the interface between LON-2::SMOC-1, as identified via ColabFold [34] and #### marks the glycosaminoglycan attachment site. Residues highlighted in yellow are those mutated to generate LON-2(mut) that cannot bind SMOC-1.
(PDF)

**S5 Fig. Multiple sequence alignment of the mature domains of DBL-1 homologs.** Clustal Omega (CLUSTAL O(1.2.4)) alignment [71] of the mature domains of DBL-1 with its homologs from other nematode species, as well as homologs from *Drosophila*, *Xenopus*, and humans. Red $ marks the residues at the interface between mature DBL-1 and SMOC-1, as identified by ColabFold [34]. The first cysteine residue in the mature domain of DBL-1 homologs is assigned as the #1 position.
(PDF)

**S6 Fig. Mutant SMOC-1 proteins are stable and detectable via western when overexpressed.** Western blot of 100 gravid adults of each indicated genotype, probed with anti-V5 (for SMOC-1) and anti-actin antibodies. The strain overexpressing WT SMOC-1::V5 is an integrated transgenic strain (*jjIs6671*). Strains overexpressing various mutant versions of SMOC-1::V5 (2 independent transgenic lines for each mutant version of SMOC-1) carry the transgenes as extra chromosomal arrays, thus the expression levels vary between different transgenic lines due to varying copy numbers of the transgene and varying degrees of stability of the transgenic arrays.
(PDF)

**S7 Fig. Endogenously tagged HA::DBL-1 is fully functional, and when over-expressed, HA::DBL-1 can cause a long body size phenotype.** (**A**) Schematic of DBL-1 protein, which contains a signal peptide (SP, dark gray), a prodomain (light gray), and a cysteine-knot containing mature domain that is the active signaling ligand (green). Inset shows residues flanking the prodomain-mature domain boundary, indicating the predicted furin cleavage sites (underlined residues) in relation to the placements of the tags (arrows). *texIs100* is an integrated transgene overexpressing a GFP-tagged DBL-1 [72]. However, when GFP was inserted in the same location as in *texIs100* in the endogenous *dbl-1* locus, the resulting allele, *jj244*, is not functional. *jj307*, *jj308*, and *jj309* are 3 identical alleles generated via CRISPR, with the HA tag inserted in the marked location. (**B**) Relative body lengths of synchronized larvae stage-matched at L4.3 vulva stage grown at 20°C, with the body length of WT worms set to 1.0. Sample sizes are 40 for each genotype. An ANOVA followed by Tukey HSD was used to test for differences between genotypes. ***$P < 0.001$; ND, no difference. (**C**) Table showing the penetrance of the Susm phenotype of double mutant strains between *sma-9(cc604)* and different *dbl-1* alleles. [a] The lack of M-derived CCs phenotype is not fully penetrant in *sma-9(cc604)* mutants. [b] Data for the 2 null *dbl-1* alleles, *nk3* and *wk7*, are from [25]. For *jj308*, 2 independent isolates were generated, and the Susm data from the 2 isolates were combined and presented in the table. Statistical analysis was conducted by comparing the strains carrying *dbl-1* alleles with the *sma-9(cc604)* single mutants. *** $P < 0.001$ (unpaired two-tailed Student's *t* test). ND, no difference. (**D**) Relative body lengths of stage-matched WT worms (set to 1.0) and worms carrying an integrated transgene (*jjIs6448*) that overexpresses HA::DBL-1. WT:

$N = 35$. *jjIs6446*, $N = 41$. ***$P < 0.001$ (ANOVA followed by Tukey HSD). (**E**) Schematic of GFP::3xFLAG::LON-2 protein in *lon-2(jj207)* animals. Original data sets are in S1 Data.
(PDF)

**S8 Fig. When overexpressed in the *smoc-1(0)* null background, SMOC-1 proteins carrying missense mutations found in *jj85* and *jj65* mutants cause a long body size phenotype.** (**A**) Diagram depicting the constructs expressing WT *smoc-1*, *smoc-1(jj85)* (pMSD47), *and smoc-1 (jj65)* (pMSD46). All plasmids contain the same 2 kb *smoc-1* promoter and 2 kb *smoc-1* 3′ UTR, as well as a 2xFLAG tag at the C terminus. Protein domains are indicated by color: navy, SP, signal peptide; green, TY, thyroglobulin-like domain; blue, EC, extracellular calcium binding domain; purple, 2xFLAG. (**B**) Relative body sizes of strains carrying indicated versions of *smoc-1* in a *smoc-1(tm7125)* null background relative to WT (set to 1.0). Gray indicates non-transgenic worms that do not express any *smoc-1*. Two strains carrying independent transgenes were measured and combined for each plasmid being tested here. Statistical analysis was done to compare transgenic strains with non-transgenic counterparts. *$P < 0.01$; ***$P < 0.001$; ND: no difference (ANOVA followed by Tukey HSD). Original data sets are in S1 Data.
(PDF)

**S1 Data. Excel files containing all the raw data for Figs 1, 3, 4, 9, 10, 11, S1, S7 and S8.** (XLSX)

**S1 Raw Images. Raw images of western blots for Figs 1, 2, 3, 5, 8, 9, 10 and S2.** (PDF)

## Acknowledgments

We thank Claire Bénard, Andy Fire, Bob Goldstein, Erik Jorgensen, Oliver Hobert, and Daniel Zinshteyn for reagents; Herong Shi for amazing technical assistance before her untimely passing; Zhiyu Liu for generating the *jj244* allele; Gwen Beacham, Erika Beyrent, and Gunther Hollopeter for sharing IP-MS and RNP CRISPR protocols; and the rest of the Liu lab for helpful discussions throughout the course of this work.

## Author Contributions

**Conceptualization:** Melisa S. DeGroot, Byron Williams, J. Christopher Fromme, Jun Liu.

**Data curation:** Melisa S. DeGroot, Byron Williams, Jun Liu.

**Formal analysis:** Melisa S. DeGroot, Byron Williams, Timothy Y. Chang, Maria L. Maas Gamboa, J. Christopher Fromme, Jun Liu.

**Funding acquisition:** Melisa S. DeGroot, Timothy Y. Chang, J. Christopher Fromme, Jun Liu.

**Investigation:** Melisa S. DeGroot, Byron Williams, Timothy Y. Chang, Maria L. Maas Gamboa, Isabel M. Larus, Garam Hong, J. Christopher Fromme, Jun Liu.

**Project administration:** Jun Liu.

**Supervision:** Jun Liu.

**Validation:** Melisa S. DeGroot, Byron Williams, Jun Liu.

**Visualization:** Melisa S. DeGroot, Byron Williams, Timothy Y. Chang, J. Christopher Fromme, Jun Liu.

**Writing – original draft:** Melisa S. DeGroot, Byron Williams, J. Christopher Fromme, Jun Liu.

**Writing – review & editing:** Melisa S. DeGroot, Byron Williams, Isabel M. Larus, J. Christopher Fromme, Jun Liu.

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
