## [Editor Report · Decision Letter 0]

20 Jan 2023

Dear Dr Liu, 

Thank you for submitting your manuscript entitled "C. elegans SMOC-1 interacts with both BMP and glypican to regulate BMP signaling" for consideration as a Research Article by PLOS Biology.

Your manuscript has now been evaluated by the PLOS Biology editorial staff as well as by an academic editor with relevant expertise and I am writing to let you know that we would like to send your submission out for external peer review.

Once your full submission is complete, your paper will undergo a series of checks in preparation for peer review. After your manuscript has passed the checks it will be sent out for review. To provide the metadata for your submission, please Login to Editorial Manager (https://www.editorialmanager.com/pbiology) within two working days, i.e. by Jan 23 2023 11:59PM.

Kind regards,

Ines

--

Ines Alvarez-Garcia, PhD

Senior Editor

PLOS Biology

---

## [Decision Letter · Decision Letter 1]

24 Mar 2023

Dear Dr Liu,

Thank you for your patience while your manuscript entitled "C. elegans SMOC-1 interacts with both BMP and glypican to regulate BMP signaling" was peer-reviewed at PLOS Biology. Please also accept my apologies for the delay in providing you with our decision. The manuscript has now been evaluated by the PLOS Biology editors, an Academic Editor with relevant expertise, and by two independent reviewers. 

The reviews are attached below. As you will see, both reviewers find the results potentially interesting, however they also mention that further experiments would be needed to strengthen the conclusions of the manuscript. The reviewers think it is essential to confirm the physical interaction between SMOC1/LON-2 and DBL-1 in developing embryos, and to show that this mechanism is conserved across species. Reviewer 1 also raises some concerns regarding the stats and thinks that a better context for the study should be provided in the Introduction.

In light of the reviews and after consulting with the Academic Editor and the rest of the team, we have decided to invite you to revise the work to thoroughly address the reviewers' reports. Given the extent of revision needed, we cannot make a decision about publication until we have seen the revised manuscript and your response to the reviewers' comments. Your revised manuscript is likely to be sent for further evaluation by all or a subset of the reviewers.

**IMPORTANT - SUBMITTING YOUR REVISION**

3. Resubmission Checklist

a) *PLOS Data Policy*

b) *Published Peer Review*

Sincerely,

Ines

--

Ines Alvarez-Garcia, PhD

Senior Editor

PLOS Biology

Reviewers' comments

Rev. 1:

The overall goal of this study is to understand the roles of conserved SMOC family proteins in BMP signaling. The experiments appear to be conducted very carefully and thoroughly and the paper is generally very well presented. The study used a nice combination of molecular genetics, mass spec, protein pull-down assays in S2 cells, and structural modeling. A key new finding is that SMOC-1 interacts with mature DBL-1 (BMP) ligand and, at least in the S2 system, may form a tripartite complex with the core of the glypican LON-2. This adds to the current knowledge about how SMOC proteins may interact with and regulate components of BMP signaling. Notably, prior studies indicate that this regulation may be complex and, in some cases, can appear paradoxical.

My overall take is that while the work is of very good quality, the authors didn't really nail down how SMOC proteins act as both positive and negative regulator of BMP signaling, a feature of SMOC proteins that was already known to some extent. This is based in part on the provided data relating to body size (and in some cases coelomocyte specification) but also to an absence of data that might directly bolster the hypothesized model. The IP pulldowns are clean, but it is a "non-physiological" heterologous assay. Still, as stated above, the study is well done, and the data might represent a sufficient advance for publishing in PLoS Biology; I would defer to the editors and those more immediately connected to the field of BMP signaling to make that call.

Specific comments:

1) Statistics and presentation. In several of the figure panels the authors use symbols instead of standard asterisks to indicate significant differences. They also state in the legend that the difference between genotypes with different symbols is p<0.001 (or P<0.0001). It may be easier for most readers to rapidly comprehend the data using asterisks and bars to indicate which comparisons are being made. In fact, Figure 3C and 4B were done this way. Also, looking at the data points (e.g., Figure 1) it is somewhat hard to imagine that jjIs5119 (with a cross) vs jjIs5798 (with a dollar sign) is different by p<0.001. Ditto tm7125 vs jj269 or jj271 in Figure S1. Furthermore, in several of the figure panels and legends (e.g., Fig1 and S1) it is stated that a t-test was used to derive p values for comparisons of phenotypic percentages. t-tests are used to compare means, so this seems incorrect.

2) Although I understand why the authors use the term "Susm penetrance" and "rescue of Susm" as a read out for SMOC-1-BMP activity, I wonder if it would be simpler for most readers to provide either the average number of posterior (M-derived) coelomocytes or to state the percentage of worms with 1-2 posterior coelomocytes. I think it can be difficult for readers who aren't geneticists to do the mental gymnastics required interpret the findings and the above presentation method may be more straightforward.

3) The apparent positive and negative roles of SMOC-1 in BMP signaling could be laid out more clearly in the introduction and throughout. It will likely be confusing to many readers as to whether SMOC-1 acts as a positive or negative regulator of BMP signaling in the coelomocyte assay, which is used but not well discussed. Some commentary about differences between these BMP pathways is warranted given that several of the pathway components, including lon-2 and sma-9, do not act in the way that one would expect in the M lineage and this gets quite confusing. For example, why does loss of lon-2 (along with smoc-1 and dbl-1) suppress the sma-9 defect? In other words, please summarize and potentially diagram what is known from previous studies to help readers frame the results. Moreover, how do the current findings apply to a model for the coelomocyte assay?

4) Overall, the protein interactions data (MS and IP) were thorough and presented well. Will the full MS dataset be included as supplementary data? What else was pulled out? Also, explaining a little more about how the double-tagged DBL-1 works would be useful. Would FLAG be bound or detected in both the mature and unprocessed forms? Is there a reason to test/show/state if SMOC-1 can bind to full length DBL-1 as was assayed for LON-2 in Fig S2?

5) In Figure 3C/D, is it necessary to show the size or Susm phenotype of siblings that didn't carry the array? To me this may add some unnecessary confusion for some readers, although I understand that it can provide a good internal control on that day. In addition, does OE of smoc-1 TY in a wild type cause a reduction in size? In other words, is there any indication of a dominant negative or antimorphic effect, particularly given some suggestion of this in Figure 4B?

6) Regarding the model, if "SMOC-1 has both a negative, LON-2-dependent role and a positive, LON-2-independent role in regulating BMP signaling", then wouldn't you expect at least some additivity in the lon-2 mutant that also overexpresses SMOC-1? This scenario wasn't included in Figure 8A, but it seems that less LON-2 should mean more diffusible DBL-1. Along those lines and in part for completeness, what is the phenotype of DBL-1 OE in a lon-2 mutant background? Regarding the statistics in 8B and the Figure legend, and as discussed above, it is stated that the difference between groups with different symbols is p <0.0001. But this seems very unlikely when comparing, for example, DBL-1 OE vs DBL-1 OC + SMOC-1 OE. Although there could a statically significant difference between these conditions, it would seem to be much more minimal based on the data point spread, and this difference is used in part to suggest a positive role for SMOC-1 in DBL-1 signaling. It also wasn't entirely clear how OE of SMOC-1(EC) stimulates BMP given that binding to DBL-1 wasn't observed in S2 system, which is presumably expressing the components at fairly high levels.

Very minor comments:

- Figure S2. Remove the word "either" from the title.

- Figure 3A did not appear to be referenced in the text.

- In Figure 3C/D, is it necessary to show the size or Susm phenotype of siblings that didn't carry the array? To me this may add some unnecessary confusion for some readers, although I understand that it can provide a good internal control on that day.

- Line 337 change vertebrates to vertebrate.

- Figure S1B. Uncapitalize smoc-1(jj276…)

Rev. 2:

Major Criticisms

1 While the finding reported regarding the tripartite interaction of SMOC1 with both glypicans and BMPs is potentially quite interesting, the results are based on predictive structural models, Co-IP experiments and inferences drawn from body size measurements following misexpression studies. Since this finding is novel and has not been reported in other model organisms, in vivo data to demonstrate the physical interaction between SMOC1/LON-2, and DBL-1 in developing embryos is needed to strengthen the study. When/where do the proposed interactions between SMOC1, LON-2 and DBL-1 occur in developing embryos?

2 As the authors intent is to generalize the interaction of SMOC1 with BMPs across species, additional data is required. Data is only presented on the predicted interaction between the mature domain of DBL-1 and C. elegans SMOC-1. Using an interaction model approach, similar to that described for cSMOC1/DBL-1, it should be possible to obtain predictive interaction data for mammalian SMOC1 and BMP 2/4. Determining whether the SMOC1/BMP interaction may exist across species, or whether it is specific to the domain arrangement of C. elegans SMOC1, is significant.

Minor Comments:

Line 23 - As C. elegans, Drosophila, and mammalian SMOC1 proteins have different domain organizations, to avoid confusion, please refer to C. elegans SMOC1 as cSMOC1 or C-SMOC1 throughout the manuscript.

Line 23/24 - As the C. elegans nomenclature for the glypican, LON-2, and the BMP, DBL-1, may not be familiar to the readership, please define LON-2 and DBL-1 and their homology to mammalian glypicans/BMPs.

Line 32 - As no data is presented to show that mammalian SMOCs can bind to BMPs, please delete the sentence "Our work provides a mechanistic basis for how the evolutionarily conserved SMOC proteins regulate BMP signaling'"

Line 38 - To be more concise I suggest rewording as follows; "BMP signaling is activated upon binding of secreted BMP ligands to complexes of the type I and type II BMP serine/threonine receptor kinases, which results in the intracellular phosphorylation of receptor-activated R-Smads".

Line 84 -Please define the zinc finger transcription factor SMA-9 and how it is relevant to BMP signaling.

Line 91 - As noted earlier, as the domain structure of C. elegans SMOC1 is different to mammalian SMOC1, I suggest rewording the sentence to "C. elegans has a single SMOC protein, cSMOC-1, which is known to regulate BMP signaling'".

Line 108 - I suggest changing the title to Generation of a C. elegans strain with a functionally-tagged SMOC-1.

Line 191 - typo - change to …. SMOCs function by competing with BMP ligands for binding to HSPGs.

---

## [Decision Letter · Decision Letter 2]

4 Jul 2023

Dear Dr Liu,

Thank you for your patience while we considered your revised manuscript entitled "C. elegans SMOC-1 interacts with both BMP and glypican to regulate BMP signaling" for publication as a Research Article at PLOS Biology. This revised version of your manuscript has been evaluated by the PLOS Biology editors, the Academic Editor and one of the original reviewers.

Based on the reviews, we are likely to accept this manuscript for publication, provided you satisfactorily address the remaining points raised by Reviewer 2 and the data and other policy-related requests stated below. Regarding the nomenclature point made by the reviewer, the Academic Editor thinks it might be better to refer to it when ambiguous to the C. elegans Smoc or worm Smoc. But you should follow C. elegans nomenclature rules.

In addition, we would like you to consider a suggestion to improve the title:

"SMOC-1 interacts with both BMP and glypican to regulate BMP signaling in C. elegans"

We expect to receive your revised manuscript within two weeks. 

*Published Peer Review History*

*Press*

Sincerely,

Ines

--

Ines Alvarez-Garcia, PhD

Senior Editor

PLOS Biology

Reviewers' comments

Rev. 2:

Responses to Major Criticisms

The authors have responded to the two major criticisms communicated in their original manuscript:

1. Although the authors were unable to present direct, in vivo, evidence for the tripartite interaction of CeSMOC-1, LON-2 and DBL-1 in C. elegans, they provide additional indirect evidence in the revised manuscript to suggest that this is possible (Figures 9 and 10 of the revised manuscript).

Reviewer Comment: While the response provides additional support for the tripartite model, the authors state that none of the possible forms of DBL-1 (mature domain, pro-domain, or full-length) were predicted to interact with LON-2/glypican. However, the authors point out that previous studies have shown that Drosophila DPP and mammalian BMP2/4 do interact with glypicans/HSPGs. As the disparity is compelling, please present additional in silico structural modeling data to compare the predicted interactions between DBL-1/LON-2 and BMP2/4/LON-2.

Are the sequences in DPP and BMP2/4 predicted to interact with HSPGs conserved in DBL-1?

2. In silico structural modeling interactions are presented for CeSMOC-1 and BMPs from Drosophila, Xenopus and human, suggesting that SMOC-BMP interactions may be evolutionarily conserved (Figure 7 of the revised manuscript).

Reviewer Comment: This response is acceptable.

Responses to Minor Criticisms

The authors now refer to C. elegans SMOC-1 as CeSMOC-1 in the introduction and discussion, but not in the results section. To avoid confusion, please change SMOC-1 to CeSMOC-1 throughout the manuscript, when referring to results obtained with CeSMOC-1.

The responses to the other minor criticisms are acceptable.

---

## [Editor Report · Decision Letter 3]

22 Jul 2023

Dear Dr Liu,

Thank you for the submission of your revised Research Article "SMOC-1 interacts with both BMP and glypican to regulate BMP signaling in C. elegans" for publication in PLOS Biology. On behalf of my colleagues and the Academic Editor, Mary Mullins, I am delighted to let you know that we can in principle accept your manuscript for publication, provided you address any remaining formatting and reporting issues. These will be detailed in an email you should receive within 2-3 business days from our colleagues in the journal operations team; no action is required from you until then. Please note that we will not be able to formally accept your manuscript and schedule it for publication until you have completed any requested changes.

PRESS

Sincerely, 

Ines

--

Ines Alvarez-Garcia, PhD

Senior Editor

PLOS Biology
